# Copper nanoparticles encapsulated in zeolitic imidazolate framework-8 as a stable and selective CO$_2$ hydrogenation catalyst

Vijay K. Velisoju [1], Jose L. Cerrillo [2], Rafia Ahmad[2], Hend Omar Mohamed[1], Yerrayya Attada[1], Qingpeng Cheng [2,3], Xueli Yao[1], Lirong Zheng[4], Osama Shekhah [3], Selvedin Telalovic[2], Javier Narciso[5], Luigi Cavallo [2], Yu Han [2,3], Mohamed Eddaoudi [3], Enrique V. Ramos-Fernández[5,6] & Pedro Castaño [1,7] ✉

Metal–organic frameworks have drawn attention as potential catalysts owing to their unique tunable surface chemistry and accessibility. However, their application in thermal catalysis has been limited because of their instability under harsh temperatures and pressures, such as the hydrogenation of CO$_2$ to methanol. Herein, we use a controlled two-step method to synthesize finely dispersed Cu on a zeolitic imidazolate framework-8 (ZIF-8). This catalyst suffers a series of transformations during the CO$_2$ hydrogenation to methanol, leading to ~14 nm Cu nanoparticles encapsulated on the Zn-based MOF that are highly active (2-fold higher methanol productivity than the commercial Cu–Zn–Al catalyst), very selective (>90%), and remarkably stable for over 150 h. In situ spectroscopy, density functional theory calculations, and kinetic results reveal the preferential adsorption sites, the preferential reaction pathways, and the reverse water gas shift reaction suppression over this catalyst. The developed material is robust, easy to synthesize, and active for CO$_2$ utilization.

Metal–metal oxide interfaces possess unique catalytic properties for various catalytic processes, and such phase boundaries have been demonstrated to be highly effective for hydrogenating CO$_2$ to methanol[1–4]. For the hydrogenation of CO$_2$ to methanol reaction, the interfaces between metallic Cu and ZnO or ZrO$_2$ are catalytically active[5–7]. However, due to the highly dynamic behavior of Cu nanoparticles under working conditions, the Cu species undergo restructuring during CO$_2$ hydrogenation, and this leads to the sintering of active phases and results in a pronounced drop in catalytic activity and selectivity[5,8–11]. Although multiple strategies have been explored to address the sintering problem (e.g., multicomponent catalysts with different structural promoters), a radically unique systematic approach is required to advance CO$_2$ hydrogenation technology[1,12–14]. Recently, metal–organic frameworks (MOFs) have been explored to solve the sintering problem by immobilizing Cu particles in the defect sites of MOFs[15–18]. Capturing Cu in the porous structure of a MOF makes these particles more resilient to restructuring, making them harder to sinter.

[1]Multiscale Reaction Engineering, KAUST Catalysis Center (KCC), King Abdullah University of Science and Technology (KAUST), Thuwal 23955-6900, Saudi Arabia. [2]KAUST Catalysis Center (KCC), King Abdullah University of Science and Technology (KAUST), Thuwal 23955-6900, Saudi Arabia. [3]King Abdullah University of Science and Technology (KAUST), Physical Sciences and Engineering Division, Advanced Membranes and Porous Materials (AMPM) Center, Thuwal 23955-6900, Saudi Arabia. [4]Beijing Synchrotron Radiation Facility, Institute of High Energy Physics, Chinese Academy of Sciences, Beijing 100049, China. [5]Laboratorio de Materiales Avanzados, Departamento de Química Inorgánica – Instituto Universitario de Materiales de Alicante, Universidad de Alicante, Apartado 99, E-03080 Alicante, Spain. [6]Advanced Catalytic Materials (ACM), KAUST Catalysis Center (KCC), KAUST, Thuwal, Saudi Arabia. [7]Chemical Engineering Program, Physical Science and Engineering (PSE) Division, KAUST, Thuwal, Saudi Arabia. ✉e-mail: pedro.castano@kaust.edu.sa

The pioneering work by Bing et al. [18]. demonstrated that MOFs are potential supports for Cu catalysts in $CO_2$ hydrogenation, opening the door for many catalytic applications. These authors immobilized $Cu/ZnO_x$ nanoparticles in the pores of UiO-bpy, a highly stable MOF with small cavities. Recently, other researchers have successfully explored Cu-containing Zr-based MOFs (UiO-66 and MOF-808) as catalysts[1,14,17]. These works demonstrated that MOFs have multiple positive effects when used as catalyst support for $CO_2$ hydrogenation.

On the one hand, the porous MOF structure can prevent or reduce Cu particle sintering, which is advantageous for preserving the catalytic function; on the other hand, structural defects in the MOF can generate a MOF-nanoparticle interface with unique catalytic properties. To date, all studies have focused on Zr-based MOFs owing to their stability and Zr promotion effect. To our knowledge, no work has been performed on immobilizing Cu nanoparticles on Zn-based MOFs to hydrogenate $CO_2$ to methanol. However, ZnO is a well-known promoter in the current industrial methanol producing catalyst[19,20]. Furthermore, a portfolio of MOFs prepared with Zn that are stable at high temperatures and pressures exists, e.g., zeolitic imidazolate frameworks (ZIFs).

We developed a robust two-step synthesis of a Cu nanoparticle containing Zn-based MOF for $CO_2$ to methanol hydrogenation catalyst that is highly active, selective, and remarkably stable. The catalyst is based on a ZIF-8 supporting MOF in which atomically dispersed $Cu^{2+}$ species are reduced to stabilized Cu sub-nanometric clusters close to $Zn^{2+}$ sites (Fig. 1a). Different catalysts based on the Cu on ZIF-8 MOF were synthesized and thoroughly characterized by different techniques such as operando spectroscopy, theoretical calculations of the

disposition of the Cu nanoparticles and adsorption capacity, and activity–stability test of $CO_2$ hydrogenation to methanol. A sustainable methanol production solution will be achievable with a complete fundamental understanding of Cu structural and electronic features in $CO_2$ hydrogenation[21].

## Results

### Synthesis of Cu species in ZIF-8

Two methods were used to integrate Cu into the ZIF-8 MOF: ion exchange (Fig. 1a, Fig. S1, S2) and conventional wet impregnation. To prepare the ion exchanged catalyst (Cu/ZIF-8|IE|), we introduced the ZIF-8 MOF in an ethanol solution of $Cu(NO_3)_2$. Then, the sample was washed in a Soxhlet with ethanol to remove the metal not strongly anchored to the MOF. In this way, we incorporated up to 12 wt.% of Cu in the MOF (Table S1). During ion exchange treatment, defects are generated in the MOF structure, and parts of linkers can be released[20]. These missing linkers can be observed using infrared spectroscopy (Fig. 1d), where the broadband centered at 1311 cm$^{-1}$ is observed because of linker deficiency. In the same figure, no peak between 1100 and 1150 cm$^{-1}$ was observed, indicating metallic defects were not present. These missing linkers were replaced via hydroxyl groups, as can be seen by IR (broad band between 3250 and 3700 cm$^{-1}$, Fig. S3). Note that Cu is anchored to these hydroxyl group ligands (Fig. 1a).

The samples were then subjected to reduction treatment at different temperatures to reduce $Cu^{2+}$ to small Cu metallic clusters and characterized using XRD (Fig. 1b). The diffractograms do not change even when the sample was reduced at 723 K. This indicates that the sample was highly stable under the reduction conditions. Only a small

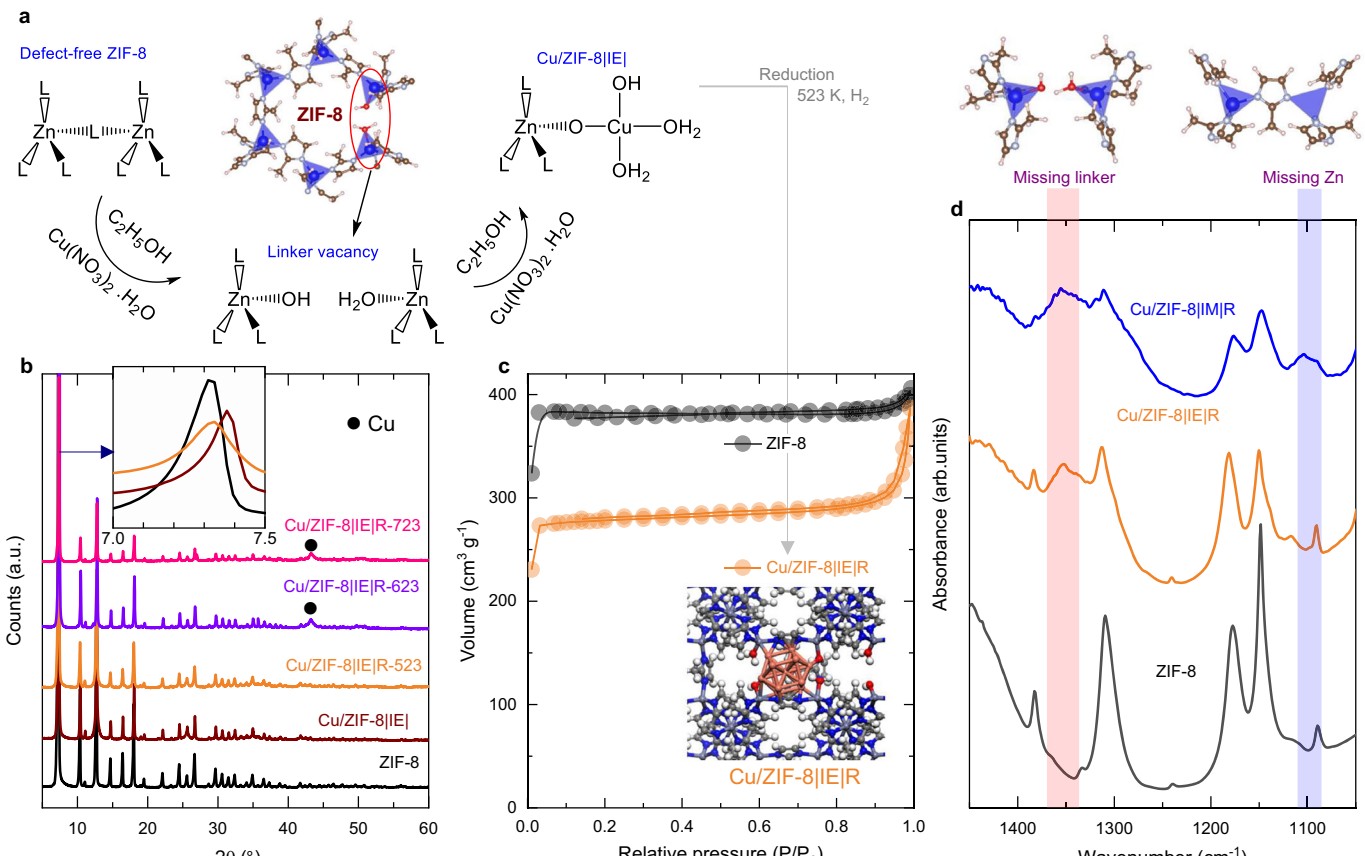

**Fig. 1 | Synthesis and structural confirmation of the catalysts. a** Cu interaction using defective nodes in ZIF-8. A linker missing in the ZIF-8 structure is replaced by −OH/OH$_2$ species that can further undergo an ion exchange to incorporate Cu into the MOF[1,20]. Powder XRD patterns of (**b**) ZIF-8 and Cu/ZIF-8|IE| samples before and after reduction at 523 K to 723 K. **c** N$_2$-adsorption isotherms of commercial ZIF-8 and the Cu/ZIF-8 | IE | R catalyst. **d** In situ DRIFT spectra of ZIF-8 before and after Cu exchange and impregnation showing linker vacancy (only) in Cu/ZIF-8 | IE | R sample, and linker and Zn vacancies in the Cu/ZIF-8 | IM | R sample.

shift of the peaks is seen before reducing the sample, thus reinforcing the idea that Cu is incorporated into the structure after reduction treatment. Cu phases were not reported after reduction at 523 K, which could be attributed to Cu particles being highly dispersed in the catalyst. Reduction at high temperatures (623 K and 723 K) showed reduced copper diffraction lines mostly due to the sintering process possible for copper when subject to high temperatures (>600 K)[22]. However, the diffraction pattern of the sample prepared by wet impregnation Cu/ZIF-8|IM| (Fig. S4) shows that the MOF degrades during the impregnation process and that the structural detriment is even more accentuated when reduced at 523 K. Peaks belonging to crystalline Cu can be seen using the wet impregnation method. It is interesting to know that the structural arrangement of the prepared Cu/ZIF-8|IE|R catalyst did not change after being stored (in a glass vial) for more than three years, explaining its long shelf-life (Fig. S5).

Thermogravimetry demonstrated the high stability of these materials (Fig. S6). Nitrogen adsorption isotherms (Fig. 1c) of the catalysts demonstrated that the exchange method developed here was essential for achieving the proper dispersion of Cu species in the ZIF-8 structure without considerably decreasing the specific surface area of the MOF. Synthesis of the same catalyst using the traditional impregnation method led to MOF degradation and a loss of all accessible porosity (no proper hysteresis is seen)[23–26].

## Structure and chemical environment of Cu species in ZIF-8

From the TEM analysis of ZIF-8 before and after Cu exchange and reduction (Fig. 2a, b, Fig. S7), we could not detect Cu nanoparticles of sufficient size that could be differentiated from that of the ZIF-8 crystals (hexagonal shaped crystals in the images). This confirms that the Cu species are highly dispersed (ultra-small or sub-nanometric level) throughout the MOF structure and provides evidence that the Cu particles are stabilized (as metallic species observed by XPS; Fig. 3e) into the ZIF structure. For comparison, the sample prepared by the traditional impregnation method demonstrated large Cu particles (Fig. S8). HAADF (Fig. 2d, e) and STEM-EDX of the ion exchange sample showed a uniform distribution of all elements (Zn, O, N and C) in the ZIF-8 structure with rhombic dodecahedron morphology.

The basicity and acidity of the catalysts were analyzed using $CO_2$ temperature programmed desorption ($CO_2$-TPD) and $NH_3$-TPD techniques. As shown in Fig. 3a, the sample prepared using the ion-exchange method after reduction (Cu/ZIF-8|IE|R) had higher basicity than ZIF-8 alone. It is well known that ZIF-8 interacts very weakly with $CO_2$. This increase in basicity after incorporating Cu may be attributed to the presence of –OH groups generated during the exchange reaction (Fig. S3 and Fig. 1a). The sample prepared via impregnation also showed higher basicity than ZIF-8. We attributed this to the decomposition of the material via reduction. Using $NH_3$-TPD (Fig. 3b), we observed an increase in the acidity of the samples with the introduction of Cu. However, this increase is much more accentuated in the case of the impregnated sample, as its decomposition leads to the formation of –OH groups, as shown in Fig. 1.

The XPS results of the catalysts are shown in Fig. 3 and Fig. S9. The Zn 2p core level spectra of all samples (Fig. 3c) revealed the presence of Zn (II) species in the catalysts with binding energy (BE) peaks at 1021.4 eV and 1044.5 eV[27]. In the Auger spectra (Zn LMM) of the ZIF-8 Cu/ZIF-8|IE|, and Cu/ZIF-8|IE|R samples (Fig. 3d), the Zn Auger region consisted of two peaks at 498 eV and 495 eV because of $Zn^{2+}$ and $Zn^0$, respectively. The presence of $Zn^0$ in both samples before or after any reduction treatment can be ascribed to photoreduction from the highly energetic X-rays used in the XPS technique and/or partial reduction of $Zn^{2+}$ species[18,28].

The Cu 2p spectrum (Fig. 3e) of the Cu/ZIF-8|IE| sample confirms the presence of oxidized $Cu^{2+}$ species with satellite peaks[29,30]. After reduction at 523 K (Cu/ZIF-8|IE|R), the CuO species were reduced to

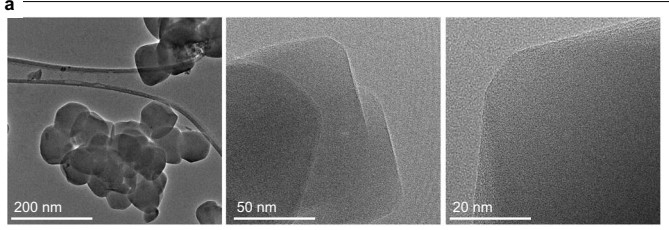

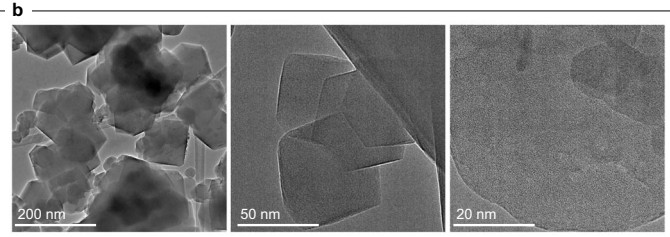

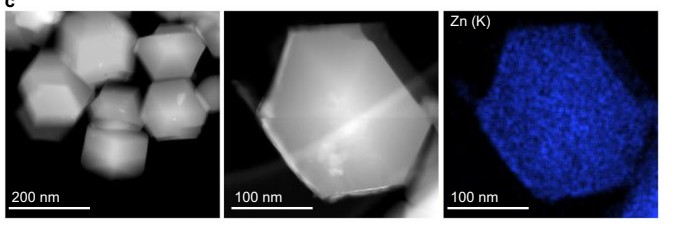

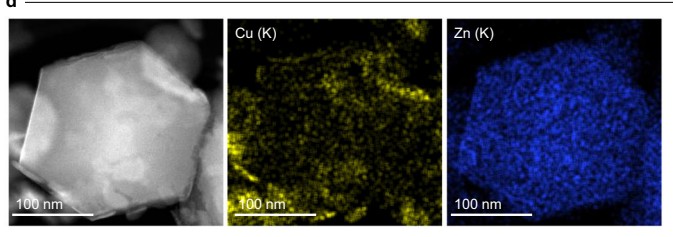

**Fig. 2 | Microscopy characterization of ZIF-8 before and after Cu introduction.** TEM images of (**a**) ZIF-8, and (**b**) Cu/ZIF-8|IE|R. HAADF-STEM-EDX images of (**c**) ZIF-8 and (**d**) Cu/ZIF-8|IE|R sample after reduction at 523 K.

metallic Cu species, as confirmed by the presence of a peak at 932 eV (Fig. 3e). The small peak is attributed to $Cu^{2+}$ from CuO that can be attributed to the outer layer of the sample from atmospheric oxygen exposure during sample preparation and transfer to the XPS apparatus[31]. The absence of a shakeup satellite peak at 947.6 eV confirms the absence of $Cu^+$ species on the surface. Compared to bulk $Cu^0$ with Cu $2p_{1/2}$ at 952.5 eV, the Cu-ion exchanged sample demonstrated a lower Cu binding energy (Cu $2p_{1/2}$ at 951.7 eV). This is likely resulted from electron injection from the conduction band of ZnO to CuO, and provides evidence for a strong interaction between Cu and Zn species[18,32]. The same was confirmed with Auger spectroscopy (Cu LMM) of these samples (Fig. 3f). No peak for $Cu^+$ was observed near 570 eV, and the peak at 568 eV confirms the presence of predominately $Cu^0$ species on the reduced catalyst[33].

The relative surface compositions obtained from this analysis demonstrated remarkably higher surface oxygen concentration after the ion exchange procedure (Table S2). After reduction, a slight decrease in surface oxygen concentration was observed. Both results were in good correlation with the in situ DRIFTS results in which an intense vibrational band due to –OH stretching was observed after Cu exchange and the subsequent reduction reaction (Fig. 4b). Note that the reduction of the sample in $H_2$ resulted in an additional O 1s peak near 533 eV because of the superficial –OH groups (Fig. S9).

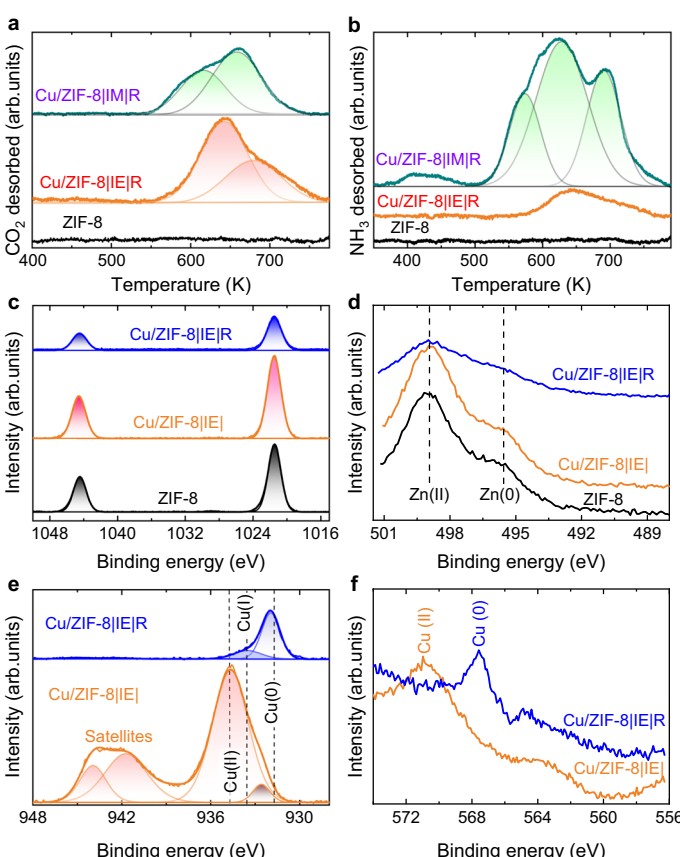

**Fig. 3 | Properties of the surface acid, base, and metal sites. a** $CO_2$-TPD, (**b**) $NH_3$-TPD profiles of ZIF-8, Cu/ZIF-8|IM|R and Cu/ZIF-8|IE|R. XPS spectra of ZIF-8 and Cu-exchanged samples: (**c**) Zn 2p, (**d**) Zn LMM; (**e**) Cu 2p, and (**f**) Cu LMM XPS spectra before (Cu/ZIF-8|IE|) and after reduction (Cu/ZIF-8|IE|R) at 523 K.

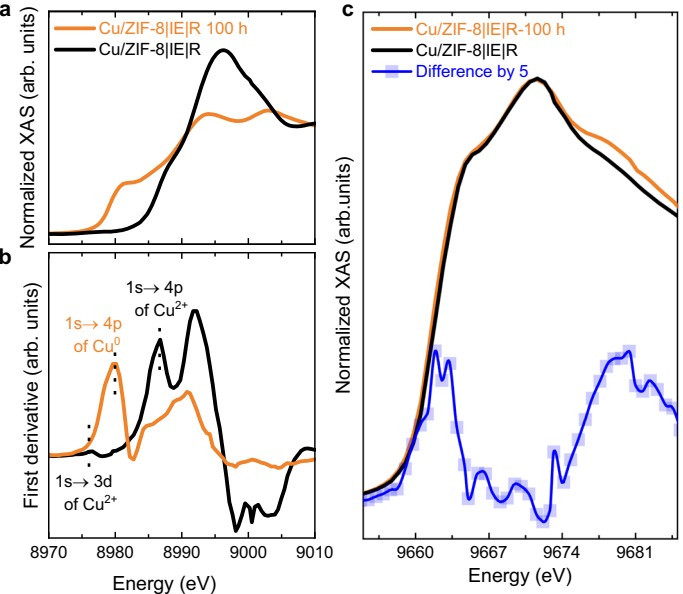

**Fig. 4 | Local environments of Cu and Zn. a** Cu K-edge XANES spectra of the freshly prepared Cu/ZIF-8|IE|R and the same catalyst after exposure to $CO_2$ reduction conditions for 100 h (Cu-ZIF-8|IE|R-100 h), (**b**) derivative spectra from both samples, and (**c**) Zn K-edge XANES spectra of the freshly prepared Cu/ZIF-8|IE|R and the same catalyst after exposure to $CO_2$ reduction conditions for 100 h. The blue line is the difference spectra multiplied by 5.

To determine the local arrangements of Cu and Zn species, XAS analysis was performed on Cu-ZIF-8|IE|R and after aging the sample in the $CO_2 + H_2$ reaction mixture under working conditions (Fig. 4a) for 100 h (Cu-ZIF-8|IE|R 100 h). The two resulting spectra are entirely different. The freshly prepared sample and the sample after $CO_2 + H_2$ treatment show the typical XANES profiles of $Cu^{2+}$ and $Cu^0$, respectively[34,35]. This is also observed in the curves of the first derivative of the XANES spectra. As shown in Fig. 4b, the derivative spectrum of the freshly prepared sample presents a weak peak at ~8977 eV for the dipole-forbidden $1s \rightarrow 3d$ transition and the main peak at ~8986.6 eV for dipole-allowed $1s \rightarrow 4p$ transition, which is characteristic of $Cu^{2+}$[36]. For the $1s \rightarrow 4p$ transition, the $Cu^{2+}$ species in the freshly prepared sample demonstrate a positive energy shift of ~1.0 eV compared to $Cu^{2+}$ in $Cu(OH)_2$, indicating that a part of the $Cu^{2+}$ species is in a distorted state[37]. The derivative spectrum of the used sample demonstrates an edge-energy feature at ~8979 eV for the $1s \rightarrow 4p$ transition, characteristic of reduced Cu[38]. All these results agree with the XPS results (Fig. 3).

For Zn K-edge XANES spectra (Fig. 4c), we see that both spectra are highly similar and correspond to the published spectra of ZIF-8. This indicates that the local environment of Zn mainly comprises $ZnN_4$[39]: Most Zn is part of the ZIF-8 structure. After subtracting the spectrum of the catalyst before using it from the sample after treatment, we see that a small peak appears at 9662 eV, indicating that the Zn white line is altered during the $CO_2$ reduction reaction. This Zn white line observation has been previously reported in other Cu/ZnO systems[34,40]. This change was attributed to the formation of interfaces between reduced Cu and partially reduced cationic zinc species[40]. These results agree with the Zn 2p XPS spectra in Fig. 3c.

## Catalytic activity, selectivity, and stability for $CO_2$ hydrogenation

The prepared catalysts were screened below 10% $CO_2$ conversion levels under 50 bar pressure (Figs. S10, S11). The principal reduction products were methanol and CO at every temperature screened. Irrespective of the different temperatures and pressures used for the Cu/ZIF-8|IE|R catalyst, we did not detect any methane production. The testing with different particle sizes (Fig. S12) confirmed no diffusion limitations for our catalyst. The synthesized Cu/ZIF-8|IE| catalyst by the ion-exchange method showed sustained activity for more than >150 h of reaction time with no significant changes in selectivity after the initial induction period of around 70 h (Fig. 5a). Depending on the exact catalyst, various induction periods were required before methanol formation to allow the catalysts to complete the necessary structural rearrangements[12,41]. Remark: The Cu/ZIF-8|IE|R synthesized by the ion exchange method required approximately 70 h for induction. Increasing the reduction time during the catalyst synthesis step did not result in any differences in terms of Cu morphology, as shown in Fig. 1b. However, after more than 70 h in reaction ($CO_2 + H_2$), Cu atoms from the lattice agglomerate into nanoparticles (Fig. 5d, e). We estimate the Cu particle size, after 100 h reaction time in the Cu/ZIF-8|IE|R-100 h catalyst, is 14 ± 3 nm (Fig. 5f and Table S3). These results agree with XRD (~15.8 nm, Table S3) and $N_2O$ pulse chemisorption (~16.6 nm, Table S3).

The Cu/ZIF-8|IE|R-100h catalyst showed >90% methanol selectivity throughout the reaction, significantly higher than the commercial Cu-Zn-Al catalyst (60%, Table S4). This means the reverse water gas shift reaction is significantly suppressed over the MOF-based catalyst compared to the commercial catalyst. At the same time, our material is almost 50% more active under similar reaction conditions (2.2 vs. 1.4 $g_{methanol}$ $g_{metal}^{-1}$ $h^{-1}$, Fig. 5b) with significantly higher TOF (0.0172 $s^{-1}$) obtained in comparison to other reported studies and commercial Cu-Zn-Al catalyst (Table S4). These key performance indicators and particularly the unparalleled stability, position of our Cu/ZIF-8|IE|R-100h catalyst as one of the top performers of its type, compared against another Cu supported on Zn-based MOF[1,14].

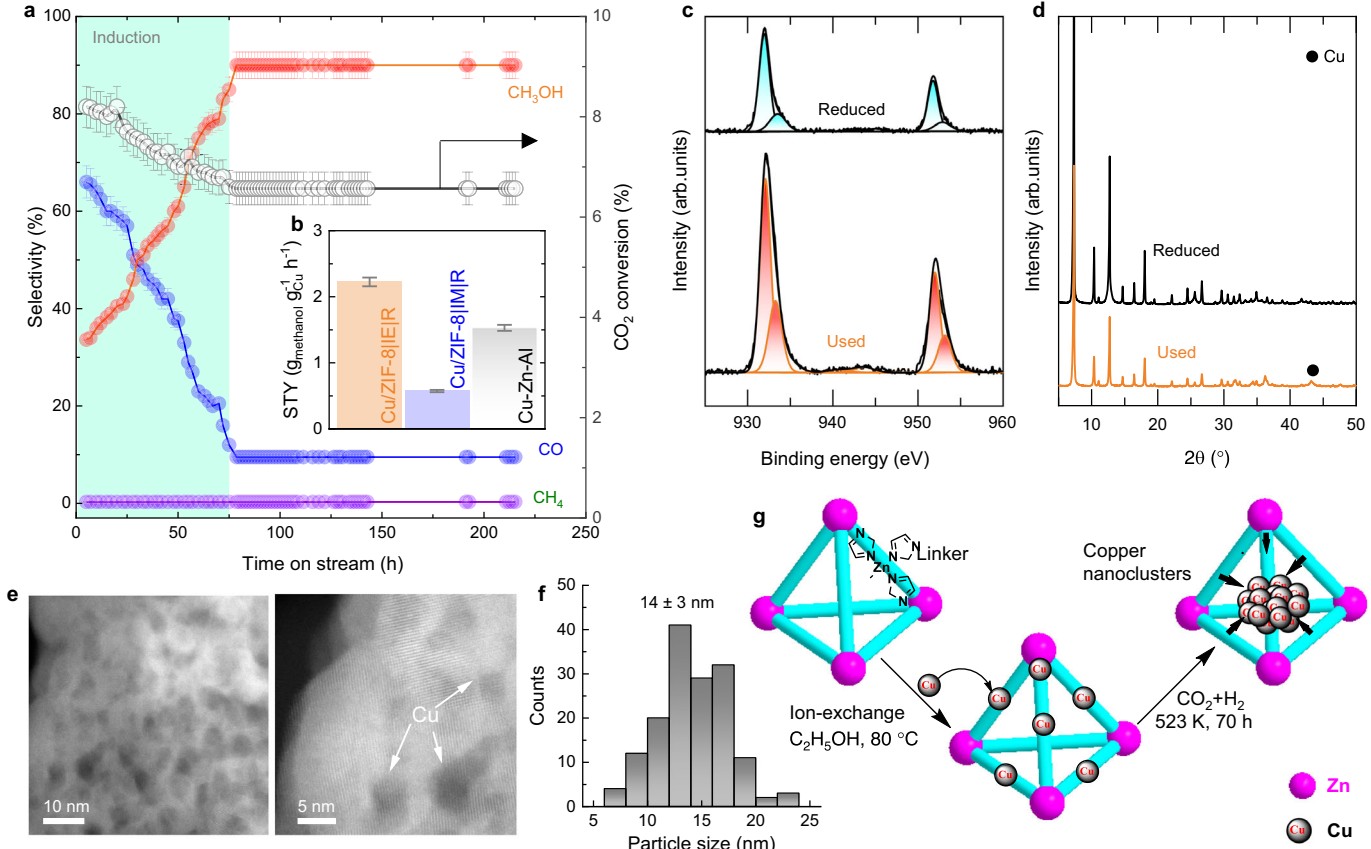

**Fig. 5 | Stability and used catalyst characterization studies. a** Time on-stream analysis of the Cu/ZIF-8 | IE | R catalyst. Reaction conditions: 80% $H_2$/20% $CO_2$ feed, T = 523 K, P = 50 bar, GHSV = 16200 $h^{-1}$. **b** Comparison of methanol space-time yields (STY) over Cu/ZIF-8 | IE | R-100 h, Cu/ZIF-8 | IM | R, and commercial Cu-Zn-Al catalyst. Reaction conditions: 80%$H_2$/20%$CO_2$ feed; T = 523 K, P = 50 bar, GHSV = 15750 $h^{-1}$. **c** Cu 2p XPS spectra of Cu/ZIF-8 | IE | R and Cu/ZIF-8 | IE | R-100 h catalysts. **d** Powder XRD patterns, (**e**) HR-TEM images of Cu/ZIF-8 | IE | R-100 h. **f** Particle size distribution of the Cu particles as obtained in the HR-TEM images for the Cu/ZIF-8 | IE | R-100 h. **g** Pictorial representation of different ZIF-8 treatment processes ultimately leading to Cu nanoparticle formation under $CO_2$ reduction conditions.

The parent ZIF-8 showed no activity (results not presented) under the experimental conditions employed. The higher selectivity of CO over the Cu/ZIF-8 | IM | R catalyst is due to the agglomeration of Cu into nanoparticles (Fig. 5g). The higher methanol selectivity observed over the Cu-ion-exchanged sample should be attributed to the presence of finely dispersed Cu nanoparticles (Fig. 5e, f), leading to a higher $CO_2$-adsorption capacity (Table S1) from the increased availability of surface hydroxyl species observed from both, XPS (Table S2) and in situ DRIFTS (Fig. 6b).

As shown in Fig. S11, the reaction temperature significantly affected the reduction product selectivity over the Cu/ZIF-8 | IE | R catalyst. The trend here correlates well with equilibrium concentrations under a similar protocol (Fig. S14). As observed in both cases, an inverse relationship is observed between methanol and CO selectivity as the reaction temperature increases from 498 to 573 K. Optimized methanol production was observed at 523 K, above which the catalyst showed more CO production with high $CO_2$ conversion. The decline in methanol selectivity at higher temperatures can be explained by the endothermic reverse water–gas shift pathway (for CO production) is thermodynamically more favorable at high temperatures, enhancing the production of CO over methanol[42,43].

The chemical environment (XPS; Fig. 5c and Fig. S15), catalyst structure (from XRD; Fig. 5d), and catalyst morphology (Fig. 5e, f) of the Cu-ZIF-8 | IE | R-100 h catalysts are compared with that of the reduced form of the catalyst (Cu-ZIF-8 | IE | R). After the reaction, the powder XRD analysis of Cu/ZIF-8 | IE | R-100 h showed a small diffraction peak at $2\theta = 43.5°$ (Fig. 5d), confirming metallic Cu presence. The surface compositions obtained from the XPS analysis (Table S2) showed a slight decrease in the surface oxygen with no changes to the Cu/Zn ratio before and after the $CO_2$ reduction reaction, confirming no loss of Cu species during $CO_2$ conversion. Therefore, after exposure to the reaction mixture (in situ reduction), further clustering of the finely dispersed Cu in the ZIF-8 MOF occur, leading to the formation of Cu nanoparticles (Fig. 5e, f) as seen in the HRTEM analysis of the catalyst (Cu-ZIF-8 | IE | R-100 h) after 100 h of time on stream.

## In situ IR for reaction mechanism

The structure and stability of the catalysts and the reaction mechanism were studied using in situ DRIFT spectroscopy at different pressures (1–25 bar) and temperatures (323–523 K; Fig. 6). ZIF-8 and Cu/ZIF-8 | IE | R-100 h share identical bands with less intense bands observed in the Cu/ZIF-8 | IM | R sample, primarily due to the structural collapse of the MOF. Additionally, a highly intense and broad vibrational band is observed in the range of 3150 to 3700 $cm^{-1}$ that can be assigned to the −O–H stretching vibrations in −OH/$OH_2$ groups, even after reduction at 523 K (Fig. 6b and Fig. S3). Exploring more deeply, we also collected the IR spectra after introducing the reaction mixture $CO_2$:$H_2$ under 25 bar pressure at the reaction temperature after in situ reduction of the sample. This resulted in new peaks between 3150 and 3700 $cm^{-1}$, mostly due to the adsorption of reactants and the interaction of $CO_2$ with the surface hydroxyl groups (Fig. 6c and Fig. S3).

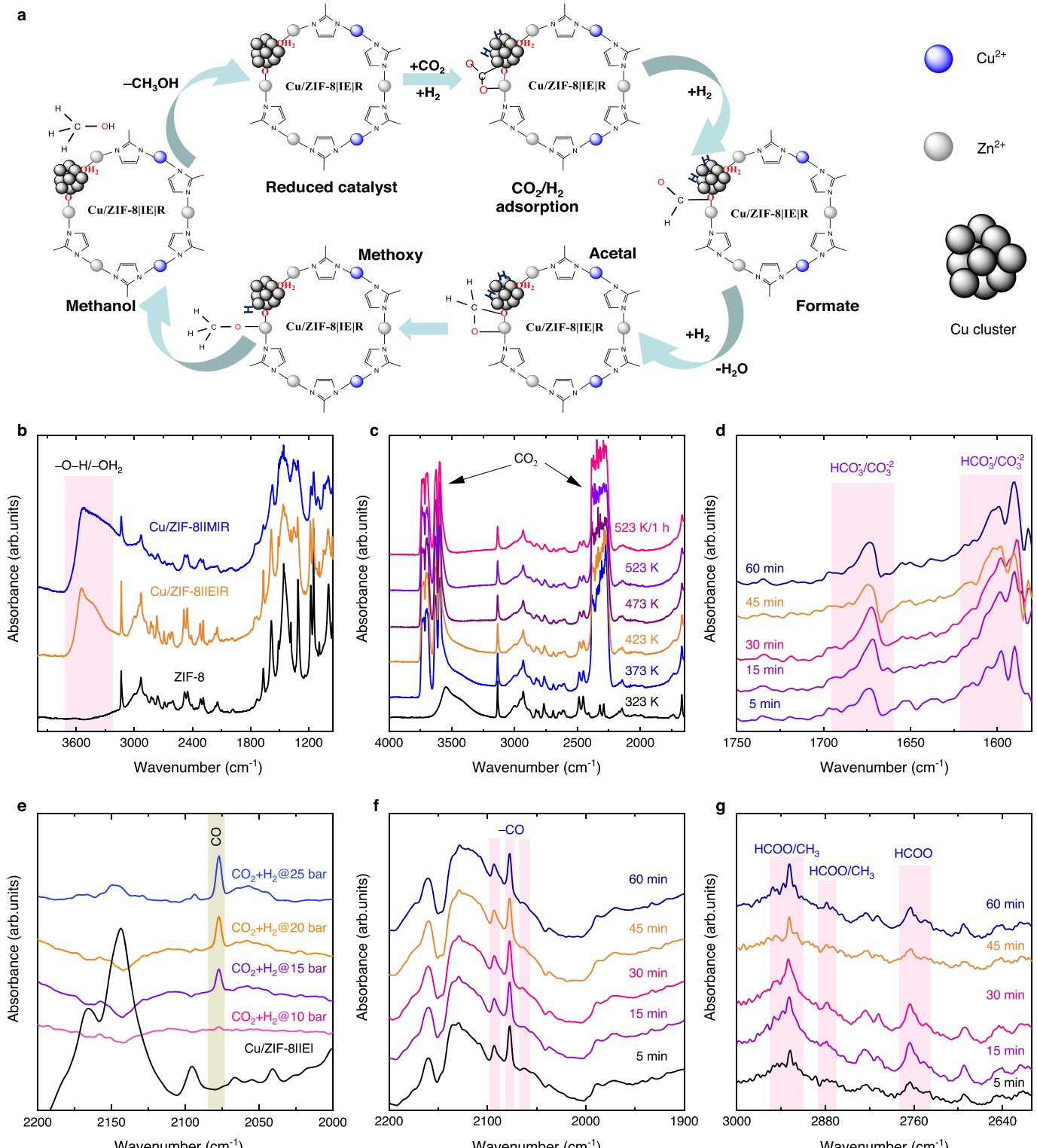

**Fig. 6 | In situ IR studies of the catalysts. a** Reaction mechanism of $CO_2$ hydrogenation to methanol through possible and identified intermediates following the formate pathway. **b** In situ DRIFT spectra of ZIF-8, Cu/ZIF-8 | IE | R, and Cu/ZIF-8 | IM | R catalysts after reduction at 523 K. **c** Cu/ZIF-8 | IE | R sample before (black) and after introducing the reaction mixture $CO_2 + H_2$ (1:3 ratio) at different temperatures under 25 bar pressure. **d** The in situ DRIFT difference spectra of Cu/ZIF-8 | IE | catalyst after introducing the reaction mixture of $CO_2$ and $H_2$ (in 1:3 ratio) in the 1750–1550 cm$^{-1}$ region, showing the formation of bicarbonate/carbonate species at 523 K under 25 bar pressure at different time intervals. The in situ DRIFT difference spectra of Cu/ZIF-8 | IE | R-100 h after reduction and introducing the reaction mixture of $CO_2$ and $H_2$ (in 1:3 ratio) in the 2200–1900 cm$^{-1}$ region, showing the formation of CO (**e**) at different time intervals and (**f**) and at different pressures at 523 K. **g** The in situ DRIFT difference spectra of Cu/ZIF-8 | IE | R-100 h catalyst after introducing the reaction mixture of $CO_2$ and $H_2$ (in 1:3 ratio) in the 3000–2600 cm$^{-1}$ region, showing the formation of formate and methoxy groups at 523 K under 25 bar pressure at different time intervals.

We analyzed the reaction mechanism by monitoring the intermediates and products from Cu/ZIF-8|IE|R-100 h in $CO_2 + H_2$ at reaction temperature (523 K) under 25 bar pressure by in situ IR spectroscopy (Fig. 6). We propose the reaction mechanism pathway as depicted in Fig. 6a based on the observed reaction intermediates. These experiments were performed at the limit allowed by the cell for safe operation, which was 25 bar (Fig. S10). The in situ IR spectra of Cu/ZIF-8|IE|R-100 h (Fig. 6b, c) exposed to the $CO_2 + H_2$ mixture at different pressures (10–25 bar) showed the formation of CO adsorbed on different Cu sites (Fig. 6e, f). The increase in the intensity of the vibrational band for CO adsorption with pressure confirms the high activity of $CO_2$ conversion at high pressures. The bands at 2093, 2078, and 2060 $cm^{-1}$ can be assigned to linearly adsorbed CO molecules on metallic Cu species[1]. We also observed methoxy species, i.e., two pairs of bands at 2960–2910 $cm^{-1}$ and 2865 $cm^{-1}$, corresponding to the $\nu(CH_3)$ and $\delta_s(CH_3)$ vibrations and bicarbonate/carbonates (Fig. 6d) in the region of 1750–1690 $cm^{-1}$ and 1680–1600 $cm^{-1}$[1,44]. The bands at 2971, 2930, 2888, and 2750 $cm^{-1}$ were assigned to the formate (HCOO) –metal species(Fig. 6g)[17,45]. Additionally, the difference IR spectra also show a gradual increase in CO vibrational band intensity with increasing pressure, confirming high conversion at higher pressures (Fig. 6f). Due to the superimposing features of methanol vibrational bands (–OH and –CH) with that of the surface hydroxyl and methoxy groups, it was challenging to monitor methanol formation in the IR cell directly. Thus, the observed HCOO⁻ species was identified as a possible key reaction intermediate in the current reaction pathway.

## $CO_2$ adsorption as a critical property

To understand the adsorption properties of $CO_2$, we performed density functional theory (DFT) calculations. Based on the results from XRD, IR, XAS, and XPS, we optimized the structure of the Cu cluster encapsulated on the ZIF-8 through linker vacancies. Specifically, we constructed a model system in which a 13-atom Cu icosahedral cluster[46] was placed near the –OH/–OH$_2$ groups of the 2-methylimidazole linker vacancy site of ZIF-8 (Fig. S17). In addition to adsorption in the absence of the Cu cluster (Fig. 7a), we considered three distinct $CO_2$ adsorption sites within this interface, namely, (1) Cu sites located away from the vacancy (Cu-distal, Figs. 7b), (2) Cu/ZnO interfacial sites or Cu sites near the –OH/–OH$_2$ groups (Cu-proximal, Figs. 7c) and (3) associated with the Zn site, which refers to the channel away from the Cu-ZnO node (Fig. 7d).

The binding of $CO_2$ to the free-standing Cu metal particle is found to be energetically weak, with a Gibbs free energy of adsorption ($\Delta G_{ad}$) of 0.25 eV at 523 K (Fig. S18). This observation is consistent with previous studies that have reported weak adsorption of $CO_2$ on Cu surfaces and clusters[47,48]. Furthermore, when $CO_2$ was exposed to bare ZIF-8 without any Cu cluster or linker vacancy (Fig. S19), no significant binding of $CO_2$ was observed. $\Delta G_{ad}$ at 523 K approximately 0 eV only (Fig. S19). This result aligns with experimental findings indicating inactivity in $CO_2$ hydrogenation. At the ligand vacancy site in the absence of a Cu nanoparticle, $CO_2$ adsorption was very weak, with a positive $\Delta G_{ad}$ value of 0.05 eV at 523 K (Fig. 7a). This is consistent with $CO_2$-TPD experiments conducted on bare ZIF-8 (Fig. 3a), where minimal $CO_2$ uptake or adsorption was observed. When examining the adsorption of $CO_2$ onto Cu sites located away from the vacancy (Cu-distal, Fig. 7b), it was observed that $CO_2$ binds in a bent state. The calculated $\Delta G_{ad}$ for this configuration was determined to be -0.28 eV.

Similarly, at the Zn-O/Cu interface (Cu-proximal, Fig. 7c), $CO_2$ exhibits a bent geometry with a $\Delta G_{ad}$ of −0.14 eV. In both cases, $CO_2$ demonstrates strong adsorption in the bent conformation, indicating the catalytic enhancement of Cu in the presence of the –OH/–OH$_2$ groups. On the other hand, the Zn-O node itself exhibits almost no binding affinity to $CO_2$. In this scenario, $CO_2$ maintains its linear geometry with the unfavorable $\Delta G_{ad}$ of 0.18 eV (Fig. 7d). Bader charge analysis[49] of the Cu$_{13}$ cluster inside the defected ZIF-8 indicates that the Cu atoms are negatively charged (Fig. S20), which explains the stronger $CO_2$ adsorption capability and the enhanced catalytic properties compared to the isolated Cu$_{13}$ cluster. Further, the Cu atoms located away from the –OH/–OH$_2$ groups exhibit more negative Bader charges compared to the Cu atoms located near the –OH/–OH$_2$ groups, which explains the stronger binding adsorption of $CO_2$ in the Cu-distal geometry.

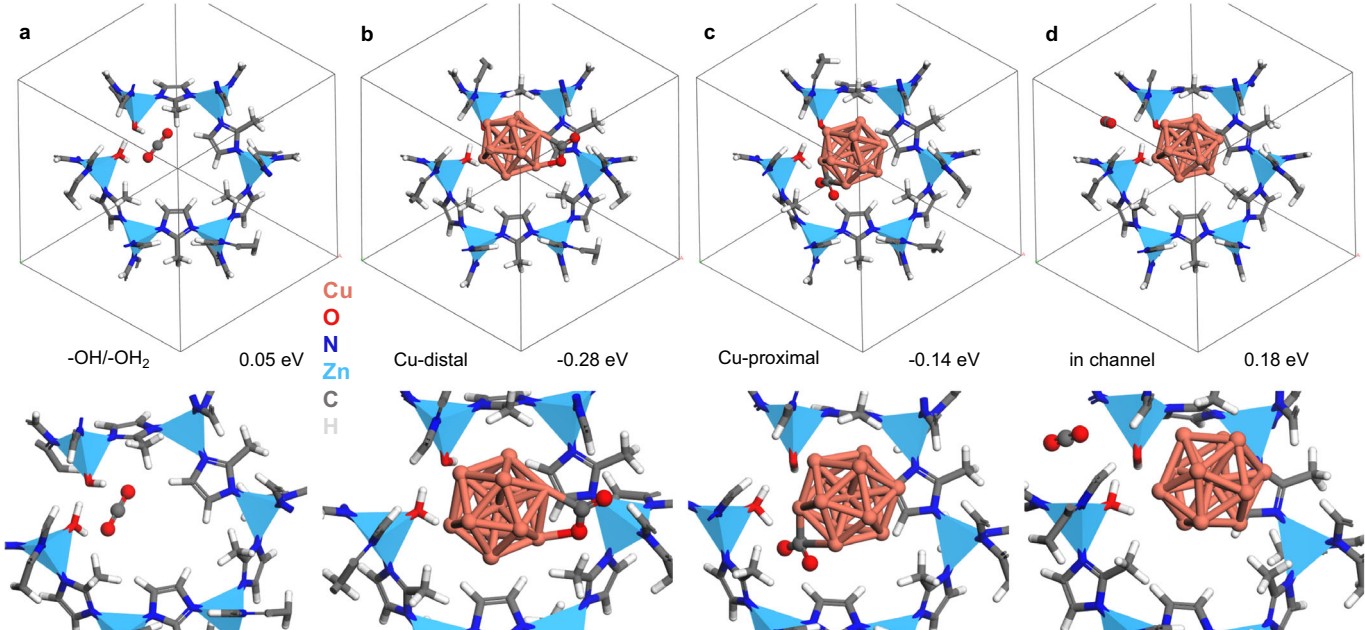

| | | | |
|---|---|---|---|
| -OH/-OH$_2$     0.05 eV | Cu-distal     -0.28 eV | Cu-proximal     -0.14 eV | in channel     0.18 eV |

Cu (brown)
O (red)
N (blue)
Zn (light-blue)
C (gray)
H (white)

**Fig. 7 | Adsorption energies of $CO_2$ at different sites in Cu/ZIF-8|IE|R catalyst.** $CO_2$ adsorbed on the following sites within defected ZIF-8: (**a**) Near the –OH/–OH$_2$ groups replacing the 2-methylimidazole linker vacancy, (**b**) Cu-distal, (**c**) Cu-proximal, and (**d**) in the channel and away from Cu cluster. A closer view of the geometries are illustrated below the respective unit-cell. The colors used to represent atoms are as follows: Cu (brown), N (blue), Zn (light-blue (tetrahedrons)), C (gray), O (red), and H (white). The numbers below each unit-cell geometry indicate the Gibbs free energy of $CO_2$ adsorption computed at 523 K and 50 bar pressure.

## Discussion

The Cu dispersed on the ZIF-8 structure prepared by a simple ion exchange method, agglomerates into ~14 nm particles after reduction and during the reaction (Fig. 5g). We show that the resulting material is highly active in the $CO_2$ hydrogenation to methanol and had outstanding stability compared with other MOF-derived catalysts. The key performance indicators of our Cu/ZIF-8|IE|R catalyst are space–time yield of 2.2 $g_{methanol}$ $g_{metal}^{-1}$ $h^{-1}$, >90% of methanol selectivity, higher than benchmark Cu−Zn−Al industrial catalyst, and >150 h stability at 523 K and 50 bar. The reason for this activity and stability are: (i) highly uniform distribution of active $Cu^0$ sites encapsulated in a Zn-based MOF that act as $CO_2$ and $H_2$ adsorption sites and (ii) the stabilization of these sites by the framework and−OH groups remaining in the catalyst after Cu reduction and the $CO_2$ to methanol reaction. The synthesis strategy employed here opens a unique pathway for chemically grafting Cu particles, and likely of other metals, that retain metal nanoparticle character at the interface while only directly replacing a fraction of the cations in the lattice. We show that the direct interaction between Zn-MOF crystals and Cu particles is mandatory for the improved catalytic reduction of $CO_2$ to methanol with the prepared Cu/ZIF-8|IE| catalysts, as no activity was observed over the bare ZIF-8. The preliminary in situ DRIFTS results for the Cu/ZIF-8|IE|R catalyst showed a significant increase of hydroxyl groups formed through linker vacancies in the ZIF-8 structure. In contrast, in situ DRIFTS analysis of the Cu/ZIF-8|IM| catalyst showed signs of both linker and Zn vacancies that led to the collapse of the MOF network. The in situ adsorption of the reaction mixture on the Cu/ZIF-8|IE| sample in the DRIFTS cell confirmed the presence of formate, which was identified as the main reaction intermediate from $CO_2$ reduction to methanol over this catalyst.

As further confirmed after DFT calculations, the constituents of the Zn−O−Cu interface formed through significant metal-support interactions are at least part of the active site that strongly adsorbs $CO_2$. The high concentration of hydroxyl groups (required for high $CO_2$ chemisorption) and the highly dispersed metallic Cu (necessary for $CO_2$ reduction) at the interface are hypothesized to explain the much higher activity of the ion exchanged catalyst produced.

The developed catalyst is exciting because it is relatively inexpensive to synthesize on a larger scale and opens different possibilities for the industrial implementation of MOFs in thermal catalysis.

## Methods

### Catalyst synthesis

The parent ZIF-8 (Basolite® or Z1200; $Zn(C_4H_5N_2)_2$ is a Zn 2-methylimidazolate material) and Cu−ZnO−$Al_2O_3$ (Cu−Zn−Al; pellets having a diameter of 5.5 mm and height of 3.6 mm) were commercially purchased from Merck (produced by BASF) and Alfa Aeser (product No. 45776), respectively. $Cu^{2+}$ exchange in ZIF-8 was performed by suspending 3 g of the commercial ZIF-8 (Basoliote® Z1200) in a solution prepared with 12.58 g of Cu nitrate hexahydrate ($Cu(NO_3)_2 \cdot 6H_2O$, purity >99%, Sigma-Aldrich) and 100 mL of ethanol. The suspension was placed in a glass bottle, secured with a screw cap, and maintained at 323 K for three days. The resulting solid was separated by filtration and washed in a Soxhlet extractor with ethanol to remove the unreacted Cu and the extracted Zn species. Then, the sample was dried in an oven at 373 K for 24 h to obtain Cu/ZIF-8|IE| . The resulting Cu/ZIF-8|IE| was then reduced to Cu/ZIF-8|IE|R in $H_2$/Ar (6%; vol/vol) at 523 K for 3 h with a ramp rate of 10 K $min^{-1}$ or aged in $CO_2 + H_2$ (1:4) reaction mixture under 50 bar pressure at 523 K for 100 h (Cu/ZIF-8|IE|−100 h) for characterization. After aging or reduction of the samples, the required pellets for the XPS and XAS analyses are prepared in a glovebox to avoid sample exposure to ambient conditions.

ZIF-8 was impregnated by Cu (in similar loading, 12% as confirmed by ICP-OES analysis) using the wetness impregnation method. ZIF-8 support was impregnated with Cu nitrate using ethanol as a solvent with constant stirring for 2 h at 300 K. The solvent was evaporated by drying at 353 K and the resulting Cu/ZIF-8|IM| was reduced following the same method as above to obtain Cu/ZIF-8|IM|R. The contents of Cu and Zn metals for all the samples were determined from ICP-OES analysis (Table S1).

### Material characterization

Powder X-ray diffraction (PXRD) analysis was done using D8 ADVANCE DaVinci (Bruker AXS) diffractometer equipped with a Bragg−Brentano geometry (Cu Kα radiation, 40 kV voltage, and tube current of 40 mA). The data was collected in a 2θ range of 10–80° at a scan speed of and a step size of 0.1°. Debye−Scherrer equation was used for the calculation of crystallite size and phase identification was done using PDF-4+ (2019) database.

$N_2$ adsorption-desorption analysis was carried out in a Micromeritics ASAP 2040 instrument to analyze the surface properties of the catalysts at 77 K. The catalyst samples were degassed at 523 K for 10 h to remove any physisorbed impurities. The specific surface areas were calculated using BET (Brunauer-Emmett-Teller) method.

Thermogravimetric analysis was carried out using a gas controller GC 200 model TGA (Mettler Toledo) device. As-synthesized sample (approximately 10 mg) was placed in an $Al_2O_3$ crucible and heated from 303 K to 1123 K with a heating rate of 10 K $min^{-1}$ in a 30 mL $min^{-1}$ flow of 6% $H_2$ in Ar (vol/vol). An empty crucible (without catalyst sample) was used as the reference material.

Metal-support interactions and reduction behaviour of the catalysts were studied using temperature-programmed $H_2$ reduction ($H_2$-TPR, Altamira AMI-200ip) with a thermal conductivity detector. Samples were pretreated in Ar at 473 K for 20 min to remove any adsorbed species. The measurements were done in a 5% $H_2$ in Ar with a total flow of 50 mL $min^{-1}$ and a heating rate of 5 K $min^{-1}$.

The $NH_3$− and $CO_2$−TPD experiments were carried out on an Autochem 2950 instrument with a TCD detector. The effluent mixture was analyzed by mass spectrometry (Hiden Analytical). The U-shaped quartz reactor was loaded with approximately 0.1 g of sample, using a heating rate of 10 K $min^{-1}$. After an in situ reduction at 523 K under 10% $H_2$/Ar for 1 h, the sample was cooled to 323 K before introducing a mixture of 10% $NH_3$ in He or 10% $CO_2$ in He for 60 min. Finally, the desorption step was carried out under pure flowing He, increasing temperature from 323 to 823 K.

The surface elemental analysis and oxidation states of all the metals/elements was analysed using K-ALPHA spectrometer (Thermo Scientific) with Al-Kα (1486.6 eV) radiation source at room temperature under ultra-high vacuum at 3 mA × 12 kV. The alpha hemispherical analyzer was operated in constant energy mode with narrow scans (for selective measurements) measuring at 50 eV and wide scans (whole energy band) at 200 eV scan pass energies. The catalyst samples were pressed, and mounted on the sample holder prior to placing in the vacuum chamber. The C 1 s peak was used at 284.5 eV to calibrate and correct the binding energy for all the elements and the charge compensation was done with the system flood gun, providing low-energy electrons. The surface composition for all the elements was estimated by integration after subtracting the background and spectrum fitting for each peak was done using a combination of Gaussian (70%) and Lorentzian (30%) lines using Avantage software.

X-ray absorption spectra (XAS) spectra of the Cu K-edge and Zn K-edge were recorded for the Cu/ZIF-8|IE|−100 h catalyst at the 1W1B station at the Beijing Synchrotron Radiation Facility (BSRF). The storage rings at the BSRF were operated at 2.5 GeV with a maximum current of 250 mA. With a Si (111) double-crystal monochromator, the data were collected in transmission mode using an ionization chamber for samples and metal foil references. All samples were pelletized as disks of 12 mm diameter with ~1 mm thickness using boron nitride powder as a binder.

A Titan Themis-Z microscope from Thermo Fisher Scientific was to perform High resolution-Transmission electron microscopy (HR-TEM) analysis for all the catalysts at an accelerating voltage of 300 kV with a beam current of 0.5 nA. Darkfield imaging was performed by scanning TEM (STEM) coupled to a high-angle annular dark-field (HAADF) detector with a convergence angle of 29.9 mrad and a HAADF inner angle of 30 mrad. Furthermore, DF-STEM imaging with an X-ray energy dispersive spectrometry (FEI SuperX, ≈0.7 sR collection angle) was used to acquire STEM-EDS spectrum-imaging data sets

ICP-OES (Varian, Inc./Agilent Model 7200-ES) was used for the quantification of all the elements in the catalysts. For microwave digestion, approximately 0.01 g of catalyst was added to a mixture of concentrated HCl (1 mL), concentrated $HNO_3$ (3 mL), and concentrated HF (1 mL) and subjected to a microwave-assisted digestion with a 15 min ramp time and 30 min hold time at 1000 W and 493 K.

$N_2O$ pulse titration for the Cu/ZIF-8 | IE | R-100 h catalyst was carried out in the Autochem-2950 (Micromeritics) unit equipped with a thermal conductivity detector to measure the copper particle size. Before the analysis, the catalyst was placed in the quartz tube after 100 h of reaction time and reduced at 523 K for 2 h in 6% $H_2$/Ar. He gas was used as carrier gas at 50 mL min$^{-1}$. The successive 5% $N_2O$/He gas doses were subsequently introduced into He stream using a calibrated injection valve at the desired temperature. The response of $N_2O$ was also monitored using a mass spectrometer (MS).

### In situ IR studies
The in situ diffuse reflectance Fourier transform infrared spectroscopy (DRIFTS) experiments were performed on a Nicolet 6700 FTIR spectrometer with a liquid-nitrogen cooled MCT detector. All the in situ experiments were carried out in aa high-temperature Harrick reaction cell with ZnSe windows equipped with a temperature programmer and connected to a gas-dosing unit with mass flow controllers (Bronkhorst). The catalyst sample (150–300μm) filled in the reaction cell was pretreated with 6% $H_2$ in Ar at 523 K for 1 h with a ramp of 10 K min$^{-1}$. The temperature decreased to 323 K, and a baseline spectrum was recorded after switching to $CO_2 + H_2$ gas. Thereafter, the sample was exposed to the same gas mixture and pressurized to 25 bar before heating from 323 to 573 K. Spectra were recorded every 5 min. The presented difference spectra were obtained by subtracting the baseline spectrum from the adsorbate spectrum. The background spectrum of dried KBr was collected in flowing He at the measurement temperature. Due to the temperature gradient in the Harrick cell, a temperature calibration at the measured surface was done (against the set temperature in the programmer), as the temperatures at the surface of the catalyst bed are generally lower than that measured at the bottom.

### Density functional theory (DFT) calculations
We utilized the Vienna ab initio Simulation Package (VASP.5.4.4)[50] for structural relaxation and electronic structure calculations. The unit cell of the 2-methylimidazole ZIF-8 (Fig. S21) was modeled with lattice parameters a = 14.736 Å, b = 14.736 Å, c = 14.736 Å, α = 109.47°, β = 109.47°, γ = 109.47°, which closely matched the experimental values[51]. To simulate a defective ZIF-8 consistent with experimental observations, we replaced a 2-methylimidazole linker with $H_2O$ and OH on the Zn atoms (Fig. S22). For the Cu-ZnO interface, we employed highly symmetrical 13-atom Cu clusters, which have been previously reported as the most energetically favorable geometry for such clusters. The Cu cluster was attached to the $-OH/-OH_2$ groups at the linker vacancy site to account for the changes in the catalytic properties of Cu in the presence of the ZnO node.

A conjugate gradient scheme was employed to relax the structures until the forces on each atom reached a magnitude of 0.01 eV Å$^{-1}$. The ionic cores were described using projector-augmented wave (PAW) pseudopotentials[50,52,53]. The Kohn-Sham one-electron valence states were expanded in the basis of plane waves with a kinetic energy cutoff of 500 eV. The exchange-correlation energy and potential were self-consistently described using the Perdew-Burke-Ernzerhof GGA functional[54], incorporating Grimme's dispersive D3 corrections[55]. For all the calculations a Mokhorst k-mesh of 3 × 3 × 3 was employed[56].

The differential Gibbs free energies of $CO_2$ adsorption ($\Delta G_{ad}$) different sites of $Cu_{13}$ encapsulated ZIF-8 were calculated using the DFT total energies, corrected by the entropic change (TΔS T is temperature and S is entropy) and the difference in zero-point energy ($\Delta E_{ZPE}$) vibrational energy ($\Delta E_{Vib}$) derived from the vibrational frequencies[57,58]:

$$\Delta G_{ad} = \Delta E_{ads} + \Delta E_{ZPE} + \Delta E_{Vib} - T\Delta S \quad (1)$$

### Catalytic tests
A multichannel highthrouput testing reactor unit (Flowrence® Avantium) comprising 4 or 16 tubular fixed-bed quartz reactors (dimensions: 2 mm ID, length 300 mm) was used for all the catalytic tests. Reactors were placed in a furnace and the flow was equally distributed over every channel using a microfluidic glass distributor. In each reactor, -0.05 g of sieved catalyst particles (150–300 μm) were loaded on 300 μL of a SiC (particle grit 40) bed to ensure the catalyst bed rested in the isothermal zone of the reactor. One reactor was always used without a catalyst as a blank. The reactors were pressurized with a mixed feed containing 20 vol.% of $CO_2$ and 80 vol.% of $H_2$ to 50 bar using a membrane-based pressure controller in the 500 to 573 K temperature range. A flow of 0.5 mL min$^{-1}$ of He was added per reactor as an internal standard. Reactor outputs were then analyzed using an Agilent 7890B gas chromatograph equipped with two sample loops, one TCD, and two FIDs detectors. The special configuration and columns of this analytical system permit us to identify and quantify $H_2$, He, CO, $CO_2$, or methanol, among other possible carbon-based products. Conversion (X, %), selectivities (S, %), and space–time yields (STY, g$_{methanol}$·g$_{metal}$$^{-1}$·h$^{-1}$) are expressed as follows:

$$X_i(\%) = \frac{F_{i,0} - F_i}{F_{i,0}} 100 \quad (2)$$

$$S_i(\%) = \frac{n_{C,i}F_i}{\sum_{Products} n_{C,i}F_i} 100 \quad (3)$$

$$STY_i\left(g_{methanol}·g_{Cu}^{-1}·h^{-1}\right) = \frac{X_{CO_2}S_i}{10000} \frac{CO_2 \, flowrate}{Cu \, weight} \quad (4)$$

Here, $n_{C,i}$ is a stoichiometric correction. Turnover frequencies (TOF) were calculated per surface metal atom. The dispersion (the ratio between copper surface atoms and total copper atoms) was evaluated using the formula:

$$D_{Cu} = \frac{6V_m}{A_m P_s} \quad (5)$$

Here, $V_m$ (7.09·10$^{21}$ nm$^3$) stands for the molar volume, and Am (4.10·10$^{22}$ nm$^2$) represents the molar area of the particles, while $P_s$ denotes the average particle size. The above equation can be simplified as:

$$D_{Cu} = \frac{1.04}{P_s} \quad (6)$$

The molar count of copper surface atoms in the catalyst ($Cu_{surf}$) was determined as:

$$Cu_{surf} = D_{Cu} \frac{W_{Cu}}{MW_{Cu}} \qquad (7)$$

Where $W_{Cu}$ is the weight of the Cu content in the catalyst, and $MW_{Cu}$ is the molecular weight of Cu. Finally, the TOF was calculated as:

$$TOF = \frac{\frac{mole\ of\ MeOH}{s}}{Cu_{surf}} \qquad (8)$$

## Data availability

The data supporting the findings of this article are available in the paper and in the Supplementary Information. Additional data are available from the corresponding author on request. Source data are provided with this paper.

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

## Acknowledgements

King Abdullah University of Science and Technology (KAUST) funded this work: BAS/1/1403. The authors acknowledge the KAUST Supercomputing Laboratory for providing high-performance computational resources and support from the KAUST Core Labs.

## Author contributions

V.K.V. and P.C. led the project and conceived the experiments. E.V.R.F. and V.K.V. designed the materials for the synthesis. V.K.V. performed and analyzed all the basic characterizations. J.L.C. performed and analyzed the catalyst testing experiments. R.A., Y.A. and L.C. performed the theoretical calculations. V.K.V. and H.O.M., performed the microscopic characterization. Q.C., X.Y., E.V.R.F., L.Z. and Y.H., performed and analyzed the X-ray absorption and XPS spectroscopy experiments. O.S., S.T., M.E. and J.N. contributed to the analysis of data. V.K.V., E.V.R.F. and P.C. wrote the manuscript with the inputs from all authors. P.C. responsible for funding acquisition.

## Competing interests

The authors declare no competing interests.

## Additional information

**Supplementary information** The online version contains Supplementary Material available at https://doi.org/10.1038/s41467-024-46388-4.

