## [Peer Review File · Nature Communications]

REVIEWER COMMENTS

Reviewer #1 (Remarks to the Author):

My research group tried extensively in loading Cu NP into Zn-MOFs for CO₂ hydrogenation but was not successful due to the instability of most of the Zn-MOFs. We also obtained results similar to the authors' Cu-ZIF-8|IM|R. It is thus a big surprise to me that the Cu-ZIF-8|IE|R sample can be so stable under the reaction condition. The key to its synthesis seems to be using EtOH at 323K. I wonder whether EtOH or its oxidation products play any role in stabilizing the framework. Overall it is an interesting piece of work. I have some other comments.

1. the authors observed Zn(0) signal on XPS even for bare ZIF-8 sample. As a result, the signal cannot be used to identify Zn(0).
2. since the Cu does not seem to be reduced in the Cu-ZIF-8|IE|R sample (only reduced after an induction period of 70 h on stream), I suggest the authors to change the symbol for avoiding confusion.
3. The average Cu cluster sizes in the catalyst can be better determined from the EXAFS data by calculating the coordination number of Cu if the author has such a data.
4. Can the authors also quantify the missing-ligand defects by determining the change of the amount of ligand in the ZIF-8?
5. From the characterizations, it is difficult to confirm the structure of Zn²⁺-Cu interface in this catalyst. I do not know if it makes sense to add back some ligand after the EtOH treatment to see if re-blocking the Zn²⁺ centers will have an effect on the activity.

Reviewer #3 (Remarks to the Author):

The work by Velisoju et al. describes a method to generate Cu-nanoclusters in a zinc-based zeolitic imidazole network. The most noteworthy findings of their investigation is the increased performance of this catalyst compared to other catalysts currently employed for the methanol synthesis, its high selectivity concerning CO as side-product of a reverse water-gas shift, and its substantial stability at operating temperatures and pressures.

This work is significant in its field, as it performs an extensive investigation of a more effective catalytic material for the methanol synthesis from CO₂, which is of crucial importance because of the immense energy and natural gas requirements of this reaction. It fits in well with the established literature about

heterogeneous, copper-based catalysts for methanol synthesis and provides a well-supported example of a promising alternative to the current industrial catalysts.

I am predominantly a theoretical chemist, so I will focus on the investigation of the catalytic material with density functional theory. I do not feel able to assess the validity of the experimental methods employed in this work in minute detail. However, as far as I can evaluate the findings with my knowledge of the field and chemistry in general, the experimental section of this work provides substantial and comprehensible evidence for the given hypotheses and conclusions and shows a thorough experimental understanding of the synthesis of the catalyst and the synthesis of methanol on it.

For the experimental part, I have three questions:

In line 213 of the submitted PDF-file, it is stated that "Cu nanoparticles agglomerate into nanoparticles". The context and figures indicate, however, that Cu-atoms dispersed in the lattice agglomerate into nanoparticles. I suggest reviewing this part and changing it accordingly.

Figure 6a shows a reaction diagram/catalytic cycle of the methanol synthesis from CO₂ and H₂ on the catalyst. However, the reaction shown there does not seem valid, as water is consumed, but not generated even if the reaction equation "CO₂ + 3H₂ -> CH₃OH + H₂O" suggests otherwise. Please review this figure and ensure that the shown data is consistent.

Lastly, the activity of the catalyst is only quantified as g(methanol)/g(cu)/h. I would suggest also converting this value into a turnover frequency (TOF) and perhaps g(methanol)/g(catalyst)/h for greater comparability with other experimental findings, as most researchers give one of these values instead to quantify catalytic performance (e.g., 10.1038/ncomms13057, 10.1021/acscatal.1c05101, and 10.1039/C8CY02546K).

As for the study of the catalyst using density functional theory, reproducibility is only partially given, and the methodology appears to be inadequate in several respects so far. The method employed in this work is VASP/PBE/500 eV with PAW potentials, but VASP, PBE, and PAW are not cited anywhere. Moreover, no k-point mesh is specified in the computational details, although this is a requirement for the periodic calculation and crucial piece of information for reproducing the results. Also, it is not clear why this functional and value for the cutoff energy were chosen, and while this is a very common level of theory, I strongly suggest backing up your choice with convergence tests or references to similar investigations (e.g., VASP/PBE-D3(BJ)/520 eV from a large-scale study applied to MOFs 10.1016/j.matt.2021.02.015).

The theoretical investigation also lacks any sort of dispersion correction (D3(BJ), D4, rVV10, or any vdW-GGA implemented in VASP) even though it covers the interaction of partially organic networks with neutral copper clusters and molecular adsorption (CO₂ on the copper cluster, for example), for which dispersion interactions are likely not negligible (see DOI:10.1039/D2CP02663E, for example). I suggest at least checking to what extent one of these dispersion correction schemes changes the results.

Lines 468-470 specify, which ions were relaxed and which were kept fixed during the geometry optimization, but upon initial inspection of two of the five investigated structures, I found that all ions were relaxed indiscriminately. Even if I misunderstood this sentence, I still suggest rephrasing the sentence for clarification.

Finally, the discussion features, along other structures, the adsorption of CO₂ on the Cu-nanocluster (Fig. 7a) and in the Cu-O-Zn interfacial site (Fig. 7d) with adsorption energies of +0.21 eV and -2.82 eV, respectively. However, CO₂ is bent and exhibits enlarged bond lengths on the Cu-nanocluster (130°, 1.23 and 1.29 Å) compared to gaseous CO₂ (180°, 1.16 Å), showing geometric signs of interaction despite a positive adsorption energy. At the same time, the molecule exhibits the same bond length and angle as gaseous CO₂ in the Cu-O-Zn-site (180°, 1.18 Å), showing no geometric signs of interaction despite a strongly negative adsorption energy. The current discussion inadequately addresses the reasons for this difference, and further investigations of, for example, the orbital interactions at play (visual or via a PDOS analysis) would be necessary to validate these findings. Moreover, there is no thermodynamic analysis of the adsorption energy, even though thermal and entropic effects have a significant impact on adsorption energies at 500-600 K. (DOI: 10.1126/sciadv.1701290, note the difference between E and G in Fig. 15)

In conclusion, the experimental section appears overall consistent and extensive enough for publication, as far as I can assess it. The theoretical section requires a rework, however, as it is neither complete nor properly reproducible in its current state. Please substantiate your choices for the functional, dispersion correction, cutoff energy, and k-point grid. Moreover, I strongly suggest a more detailed investigation of the interaction of CO₂ with the catalyst in the calculated structures. Additionally, I suggest calculating/discussing the influence of thermodynamic contributions to the given adsorption energies.

Reviewer #2 (Remarks to the Author):

The manuscript "Copper nanoparticles encapsulated in zeolitic imidazolate framework-8 as a highly stable and selective catalyst for CO₂ hydrogenation to methanol", submitted by Velisoju et al, describes a controlled two-step method to synthesize finely dispersed Cu on a zeolitic imidazolate framework-8 (ZIF-8).

Although Cu-based catalysts for methanol synthesis is a century old, years of intense researches have aimed to determine the mechanism and active sites. The catalytic

nature of the Zn in the catalyst is still highly debated, and the mechanism is not yet fully experimentally verified. The authors report that this catalyst suffers a series of transformations during the CO₂ hydrogenation to methanol, leading to ~14 nm Cu nanoparticles encapsulated on the Zn-based MOF that are highly active (2-fold higher 31 methanol productivity than the commercial Cu–Zn–Al catalyst), very selective (> 90%), and remarkably stability. However, the structure-activity relationship is unclear, and the characterizations are difficult to support the final conclusion. To date, the confinement of metal active phase is common means to improve catalytic activity. This reviewer recommends the rejection of the work for publication by Nature Communications.

1. It's not rigorous that the author proposed that "The samples were then subjected to reduction treatment at different temperatures to reduce Cu²⁺ to small Cu metallic clusters and characterized using XRD (Fig. 1b)." (lines 89, pages 4) The author should mark the characteristic peaks of zero-valent copper. In addition, the author described that "Cu phases were not reported even after reduction at 723 K, which could be attributed to the fact that Cu particles are highly dispersed in the catalyst."(lines 93, pages 5) In my opinion, the CuO diffraction peak can be observed at 43°(2 theta) under 623K H₂ reduction.

2. It is not trivial to detect the reduction of ZnO to Zn by XPS (Angew. Chem. Int. Ed. 2014, 53, 5941;J. Colloid Interface Sci. 2009, 338, 16). It's hardly to observed that some of the Zn²⁺ sites might have been reduced during hydrogenation form the result of Figure 3

3. The author described that the sample are exposed in atmospheric oxygen before transfer to the apparatus during XPS processes (lines 153, pages 8). This will cause the valence state to be inaccurate.

4. The author described that "The absence of a shakeup satellite peak at 947.6 eV confirms the absence of Cu⁺ species on the surface" (lines 154, pages 8) This is completely wrong because of the shakeup satellite peaks of Cu⁺ species are very weak and difficult to distinguish.

5. The author thought all XANES results agree with the XPS results. As we known, the XPS spectra is reflected the surface information, while the XANES results mainly embodies the bulk structure.

6. According to Fig. 5, the catalyst has an obvious induction period. The change of catalyst structure during induction period needs to be studied emphatically.

Reviewer #1:

Comment #1.1: My research group tried extensively in loading Cu NP into Zn-MOFs for CO₂ hydrogenation but was not successful due to the instability of most of the Zn-MOFs. We also obtained results similar to the authors' Cu-ZIF-8|IM|R. It is thus a big surprise to me that the Cu-ZIF-8|IE|R sample can be so stable under the reaction condition. The key to its synthesis seems to be using EtOH at 323 K. I wonder whether EtOH or its oxidation products play any role in stabilizing the framework. Overall it is an interesting piece of work. I have some other comments.

§ Response 1.1. We thank the reviewer for his valuable time and comments in assessing our work and commenting on the novelty. We have made the necessary changes in the revised manuscript as per the comments.

Comment #1.2: The authors observed Zn(0) signal on XPS even for bare ZIF-8 sample. As a result, the signal cannot be used to identify Zn(0).

§ Response 1.2. We completely agree, so we have corrected this part in the revised manuscript on p 7.

Comment #1.3: Since the Cu does not seem to be reduced in the Cu-ZIF-8|IE|R sample (only reduced after an induction period of 70 h on stream), I suggest the authors to change the symbol for avoiding confusion.

§ Response 1.3. We have changed this term throughout the revised manuscript.

Comment #1.4: The average Cu cluster sizes in the catalyst can be better determined from the EXAFS data by calculating the coordination number of Cu if the author has such a data.

§ Response 1.4. We carefully analyzed Cu particle sizes using three different techniques and found a good consistency between each other (Table S3): an average size of around 14 nm (Fig. Sf and Fig. S13). We analyzed the EXAFS data again, and the error in the fitting to calculate the Cu coordination was significant. That means the coordination number of Cu calculated from the EXAFS data is unreliable. Based on the Cu particle size distribution, it is estimated that the number of atoms in each particle is in the order of hundreds (a Cu cluster of ~0.7 nm has an atomicity of 55¹).

Comment #1.5: Can the authors also quantify the missing-ligand defects by determining the change of the amount of ligand in the ZIF-8?

§ Response 1.5. Quantifying the missing linkers in the defective MOF is challenging. Cirujano and Llabrés i Xamena² developed a method to calculate the MOF organic content by measuring the ash weight after combustion in a thermobalance. By comparing the theoretic organic matter with that obtained from this method, these authors determined the number of missing linkers in defective samples. This method is impractical in our case for two reasons: (1) copper deposition on the MOF occurs during ethanol exchange, leading to the formation of hydroxyl groups; (2) Zn²⁺ can undergo reduction and sublimation during heat treatment. Consequently, the thermogravimetric method described before cannot be used for determining the change in the amount of ligand in the ZIF-8. We detected linker vacancies using IR spectroscopy, although it is not a quantitative technique.²

Comment #1.6: From the characterizations, it is difficult to confirm the structure of the Zn²⁺-Cu interface in this catalyst. I do not know if it makes sense to add back some ligand after the EtOH treatment to see if re-blocking the Zn²⁺ centers will have an effect on the activity.

§ Response 1.6. In our catalyst preparation methodology, the synthesis occurs in a single step, where linker loss and Cu introduction into the structure happen simultaneously, as illustrated in Figure 1 of the manuscript. These processes cannot be decoupled. If we were to develop a method to incorporate the linker separately, it would be challenging to determine its specific placement on Zn²⁺ or other species like Cu cations in the structure. Additionally, the material exhibits dynamic behavior under reaction conditions, requiring approximately ~70 h to attain stability in both conversion and selectivity. All spectroscopic characterization techniques (TEM, XAS, XPS, and XRD) and theoretical calculations (in our work and others from the literature) consistently indicate the significance of the Cu-Zn²⁺ interaction, which plays a pivotal role in achieving excellent catalytic results. In particular, the *in situ* DRIFT results clearly illustrate the nature of the species formed and underscore the importance of the Cu-Zn²⁺ interface for catalytic activity.

Reviewer #2:

Comment #2.1: The manuscript "Copper nanoparticles encapsulated in zeolitic imidazolate framework-8 as a highly stable and selective catalyst for CO₂ hydrogenation to methanol", submitted by Velisoju et al, describes a controlled two-step method to synthesize finely dispersed Cu on a zeolitic imidazolate framework-8 (ZIF-8). Although Cu-based catalysts for methanol synthesis is a century old, years of intense researches have aimed to determine the mechanism and active sites. The catalytic nature of the Zn in the catalyst is still highly debated, and the mechanism is not yet fully experimentally verified. The authors report that this catalyst suffers a series of transformations during the CO₂ hydrogenation to methanol, leading to ~14 nm Cu nanoparticles encapsulated on the Zn-based MOF that are highly active (2-fold higher methanol productivity than the commercial Cu-Zn-Al catalyst), very selective (>90%), and remarkably stability. However, the structure-activity relationship is unclear, and the characterizations are difficult to support the final conclusion. To date, the confinement of metal active phase is common means to improve catalytic activity. This reviewer recommends the rejection of the work for publication by Nature Communications.

§ Response 2.1. The number of works delving with Cu/ZnO-based catalysts for methanol synthesis is extensive. The reaction mechanism in this type of reaction is not fully understood due to the lack of characterization tools under process conditions and the dynamic nature of the catalysts. The importance of the Cu-Zn interface is well-established. Maximizing this interface can significantly enhance activity. We know of previous attempts to achieve this goal, such as using Cu nanoparticles or supporting Cu atomically dispersed in ZnO. However, catalyst deactivation can often occur as particles grow during the reaction. The emergence of MOFs in catalysis has provided new opportunities to enhance this interaction. In our work, we utilized a Zn-MOF for the first time to stabilize Cu particles near Zn catalysts within the MOFs without collapsing their structural integrity. This approach ensures both stability and activity in the final material. The novelty of our article lies in the enhanced catalytic activity, selectivity, and stability produced by the interaction of the Cu particles with the Zn of the ZIF-8 structure, which means that once the steady state is reached, both the activity and selectivity remain stable.³ We have made significant changes in the revised manuscript to clarify the structure-activity and the effect of confinement.

Comment #2.2: It's not rigorous that the author proposed that "The samples were then subjected to reduction treatment at different temperatures to reduce Cu²⁺ to small Cu metallic clusters and characterized using XRD (Fig. 1b)." (lines 89, pages 4) The author should mark the characteristic peaks of zero-valent copper. In addition, the author described that "Cu phases were not reported even after reduction at 723 K, which could be attributed to the

fact that Cu particles are highly dispersed in the catalyst.”(lines 93, pages 5) In my opinion, the Cu⁰ diffraction peak can be observed at 43° (2 theta) under 623 K H₂ reduction.

§ Response 2.2. We have marked the diffraction line at $2\theta = 43^\circ$ due to metallic Cu observed only in the case of Cu-ZIF-8|IE|R reduced at 623 K and 723 K. This is expected given that the sintering process is inevitable for Cu species when subjected to temperatures above 600 K.⁴ We have used this data to check the thermal stability of ZIF-8 MOF after copper exchange and to corroborate our experimental evidence from TGA analysis. Further, our Cu-ZIF-8|IE|R catalyst reduced at 523 K did not show diffraction lines of Cu⁰ due to its fine dispersion, as shown in STEM-EDX analysis. We revised this discussion for more clarity to reflect these changes in the revised manuscript on p 5.

Comment #2.3: It is not trivial to detect the reduction of ZnO to Zn by XPS (Angew. Chem. Int. Ed. 2014, 53, 5941; J. Colloid Interface Sci. 2009, 338, 16). It's hardly to observe that some of the Zn²⁺ sites might have been reduced during hydrogenation form the result of Figure 3.

§ Response 2.3. This is true. We have amended this part in the revised manuscript on p 7.

Comment #2.4: The author described that the sample are exposed in atmospheric oxygen before transfer to the apparatus during XPS processes (lines 153, p 8). This will cause the valence state to be inaccurate.

§ Response 2.4. The sample exposure to the atmosphere will cause metallic oxidation. Thus, we curtailed this effect by handling the sample preparation for XPS analysis in a glovebox. We have also analyzed the auger spectra for both the elements (Cu and Zn) to confirm the oxidation states. In addition, an identical protocol was also used for the XAFS analysis that indicated the presence of metallic copper species (and the absence of oxidated ones), corroborating the XPS results.

Comment #2.5: The author described that “The absence of a shakeup satellite peak at 947.6 eV confirms the absence of Cu⁺ species on the surface” (lines 154, pages 8) This is completely wrong because of the shakeup satellite peaks of Cu⁺ species are very weak and difficult to distinguish.

§ Response 2.5. We further provide the auger (Cu LMM) spectra and XAS analyses to support this argument in Fig. 3f and discussions on p 8.

Comment #2.6: The author thought all XANES results agree with the XPS results. As we known, the XPS spectra is reflected the surface information, while the XANES results mainly embodies the bulk structure.

§ Response 2.6. We fully concur with the observation regarding the surface sensitivity of XPS and the bulk nature of XAS. Indeed, each technique differs in the penetration depth: XPS~ 10 nm and XANES~ 100-1000 nm. It is crucial to emphasize that in MOF materials like ZIF-8, the photoelectrons generated by X-rays in XPS have a limited mean free path of approximately 10 nm. Considering that ZIF-8 crystals possess an average crystal size of 100 nm, our analysis covers a substantial portion of the crystal volume.^{5,6} We employed HR-TEM and EDX analyses, supporting our findings. These techniques revealed that the external part of the crystals does not exhibit a different composition from the internal part, indicating a high level of homogeneity within the samples. Therefore, it is unsurprising that both XPS and XAS yield consistent results.

Comment #2.7: Fig. 5 shows the catalyst has an obvious induction period. The change of catalyst structure during induction period needs to be studied emphatically.

§ Response 2.7. Our study focuses on characterizing the state of the catalyst after 70 h of reaction. While we agree that what occurs during the induction period is interesting, it falls beyond the scope of this work. We aim to comprehensively understand the catalyst behavior and performance during the stabilized active phase.

Reviewer #3:

Comment #3.1: The work by Velisoju et al. describes a method to generate Cu-nanoclusters in a zinc-based zeolitic imidazole network. The most noteworthy findings of their investigation are the increased performance of this catalyst compared to other catalysts currently employed for methanol synthesis, its high selectivity concerning CO as a side-product of a reverse water-gas shift, and its substantial stability at operating temperatures and pressures. This work is significant in its field, as it extensively investigates a more effective catalytic material for the methanol synthesis from CO₂, which is of crucial importance because of the immense energy and natural gas requirements of this reaction. It fits well with the established literature about heterogeneous, copper-based catalysts for methanol synthesis and provides a well-supported example of a promising alternative to the current industrial catalysts.

I am predominantly a theoretical chemist, so I will focus on investigating the catalytic material with density functional theory. I do not feel able to assess the validity of the experimental methods employed in this work in minute detail. However, as far as I can evaluate the findings with my knowledge of the field and chemistry in general, the experimental section of this work provides substantial and comprehensible evidence for the given hypotheses and conclusions and shows a thorough experimental understanding of the synthesis of the catalyst and the synthesis of methanol on it.

§ Response 3.1. We thank the reviewer again for his valuable time invested in our work and for kindly criticizing our modeling section. Following the advice, we have involved a group of experts in modeling (new authors) and made the necessary calculations and changes in the revised manuscript.

Comment #3.2: For the experimental part, I have three questions: In line 213 of the submitted PDF-file, it is stated that "Cu nanoparticles agglomerate into nanoparticles". However, the context and figures indicate that Cu-atoms dispersed in the lattice agglomerate into nanoparticles. I suggest reviewing this part and changing it accordingly.

§ Response 3.2. We agree, so we have revised this part in the manuscript on p 11.

Comment #3.3: Figure 6a shows a reaction diagram/catalytic cycle of the methanol synthesis from CO₂ and H₂ on the catalyst. However, the reaction shown there does not seem valid, as water is consumed but not generated even if the reaction equation "CO₂ + 3H₂ → CH₃OH + H₂O" suggests otherwise. Please review this figure and ensure that the shown data is consistent.

§ Response 3.3. We have fixed this error in Fig. 6a in the revised submission on p 15.

Comment #3.4: Lastly, the activity of the catalyst is only quantified as g(methanol)/g(cu)/h. I would suggest also converting this value into a turnover frequency (TOF) and perhaps g(methanol)/g(catalyst)/h for greater comparability with other experimental findings, as most researchers give one of these values instead to quantify catalytic performance (e.g., 10.1038/ncomms13057, 10.1021/acscatal.1c05101, and 10.1039/C8CY02546K).

§ Response 3.4. We thank the reviewer for the suggested references to calculate the TOF.⁷⁻⁹ Considering the recommendation, we compared the data presented in Table S4. The outcomes reveal that our catalyst significantly surpasses the reported TOFs for other MOFs and Cu-Zn-Al catalysts. Remarkably, our catalyst TOF falls within the same order of magnitude as those mentioned in the literature that assert their position among the most active ones. To provide comprehensive information, we have integrated a fresh column into

Table S4, showcasing the TOFs, and provided relevant discussion in the manuscript on p 11. As metal loading plays a crucial role (limited to a certain extent for exchange in MOFs and key to higher methanol productivity), we report both comparisons: TOF (s^{-1})⁷⁻⁹ and STY ($\text{g}_{\text{methanol}} \text{g}_{\text{Cu}}^{-1} \text{h}^{-1}$).^{1,10,11} Furthermore, in the experimental section, we have detailed the methodology for calculating these values on p 24.

Comment #3.5: As for the study of the catalyst using density functional theory, reproducibility is only partially given, and the methodology appears inadequate in several respects. The method employed in this work is VASP/PBE/500 eV with PAW potentials, but VASP, PBE, and PAW are not cited anywhere. Moreover, no k-point mesh is specified in the computational details, although this is a requirement for the periodic calculation and crucial piece of information for reproducing the results. Also, it is not clear why this functional and value for the cutoff energy were chosen, and while this is a very common level of theory, I strongly suggest backing up your choice with convergence tests or references to similar investigations (e.g., VASP/PBE-D₃(BJ)/520 eV from a large-scale study applied to MOFs 10.1016/j.matt.2021.02.015).

§ Response 3.5. We have incorporated the references for VASP¹²⁻¹⁴, PBE¹⁵ and PAW^{13,14}, and provided the k-point mesh (3x3x3) in the DFT methods section in the revised manuscript on p 23. We agree with the reviewer that the methodology selected was to optimize calculation time and accuracy. We believe an energy cut-off of 500 eV is accurate enough.^{16,17} In order to test this, we recalculate the electronic energies of selected structures with a cut-off energy of 520 eV. We find that the absolute values of adsorption energies are stabilized by 0.03 eV (minor change) but the trend remains exactly the same. Therefore, we report all the calculations done by VASP/PBE-D₃(BJ)/500 eV.

Comment #3.6: The theoretical investigation also lacks any sort of dispersion correction (D₃(BJ), D4, rVV10, or any vdW-GGA implemented in VASP) even though it covers the interaction of partially organic networks with neutral copper clusters and molecular adsorption (CO₂ on the copper cluster, for example), for which dispersion interactions are likely not negligible (see DOI:10.1039/D2CP02663E, for example). I suggest at least checking to what extent one of these dispersion correction schemes changes the results.

§ Response 3.6. We thank the reviewer again for his constructive criticism. We have incorporated dispersion corrections (DFT-D₃) in the new calculations, as detailed in the computational details of the revised manuscript on p 23.

Comment #3.7: Lines 468-470 specify, which ions were relaxed and which were kept fixed during the geometry optimization, but upon initial inspection of two of the five investigated structures, I found that all ions were relaxed indiscriminately. Even if I misunderstood this sentence, I still suggest rephrasing the sentence for clarification.

§ Response 3.7. No ions were frozen, and the full unit cell was allowed to relax to optimize the geometries. However, to conduct the frequency calculations economically, we freeze all the atoms except the adsorbate to arrive at Gibbs free energies of adsorption.^{18,19}

Comment #3.8: Finally, the discussion features, along other structures, the adsorption of CO₂ on the Cu-nanocluster (Fig. 7a) and in the Cu-O-Zn interfacial site (Fig. 7d) with adsorption energies of +0.21 eV and -2.82 eV, respectively. However, CO₂ is bent and exhibits enlarged bond lengths on the Cu-nanocluster (130°, 1.23 and 1.29 Å) compared to gaseous CO₂ (180°, 1.16 Å), showing geometric signs of interaction despite a positive adsorption energy. At the same time, the molecule exhibits the same bond length and angle as gaseous CO₂ in the Cu-O-Zn-site (180°, 1.18 Å), showing no geometric signs of interaction despite a strongly negative adsorption energy. The current discussion inadequately addresses the reasons for this difference, and further investigations of, for example, the orbital interactions at play (visual or via a PDOS analysis) would be necessary to validate these findings. Moreover, there is no thermodynamic analysis of the adsorption energy, even though thermal and entropic effects have a significant

impact on adsorption energies at 500-600 K. (DOI: 10.1126/sciadv.1701290, note the difference between E and G in Fig. 15).

§ Response 3.8. We agree with the reviewer on the discrepancy of adsorbed geometry and adsorption energy values. We reviewed our calculations from the beginning by optimizing and relaxing each structure completely with a finer k-points mesh of 3x3x3. The Cu cluster indeed binds CO₂ in a bent state consistent with the literature^{20,21} with a Gibbs free energy of adsorption -0.28 eV at experimental conditions (523 K and 50 bar). Moreover, the strongest binding corresponds to the bent geometries of CO₂, which also indicates better activation of CO₂ and in turn implies better CO₂RR catalytic activity of the Cu encapsulated ZIF-8 compared to pure ZIF-8. We agree that the Gibbs free energy of adsorption gives a better thermodynamic picture of CO₂ adsorption. We correct the electronic adsorption energies (ΔE_{elec}) to free energy of adsorption (ΔG) using the equation:

$$\Delta G = \Delta E_{\text{elec}} + E_{\text{ZPE}} + E_{\text{Vib}} - T\Delta S \quad (1)$$

where, E_{ZPE} is the zero-point energy calculated by displacement method with normal mode frequencies, E_{Vib} is the vibrational energy at 523K, and ΔS is the entropy change. We report these values in the new Figure 7 of the revised manuscript. Furthermore, following the suggestion to analyze the adsorption energy trend at different sites from the electronic structure perspective we calculate the Bader charges on the Cu13 cluster when grafted on the ZnO node (See Figure below, included in the SI of the revised manuscript). We observe that the Cu atoms away from the vacancy are more negatively charged compared to the Cu atoms near the vacancy. The more negatively charged Cu atoms away from the vacancy could act like more basic Lewis sites giving stronger CO₂ adsorption.

Comment #3.9: In conclusion, the experimental section appears overall consistent and extensive enough for publication, as far as I can assess it. The theoretical section requires a rework, however, as it is neither complete nor properly reproducible in its current state. Please substantiate your choices for the functional, dispersion correction, cutoff energy, and k-point grid. Moreover, I strongly suggest a more detailed investigation of the interaction of CO₂ with the catalyst in the calculated structures. Additionally, I suggest calculating/discussing the influence of thermodynamic contributions to the given adsorption energies.

§ Response 3.9. We have addressed all the concerns raised and modified the manuscript accordingly. We thank the reviewer again for the criticism, which allowed us to amend some inaccuracies in the originally submitted manuscript.

References

1. Zhu, Y. *et al.* Copper-zirconia interfaces in UiO-66 enable selective catalytic hydrogenation of CO₂ to methanol. *Nat. Commun.* 2020 11 11, 1–11 (2020).
2. Cirujano, F. G. & Llabrés I Xamena, F. X. Tuning the Catalytic Properties of UiO-66 Metal-Organic Frameworks: From Lewis to Defect-Induced Brønsted Acidity. *J. Phys. Chem. Lett.* **11**, 4879–4890 (2020).
3. Lam, E., Noh, G., Larmier, K., Safonova, O. V. & Copéret, C. CO₂ hydrogenation on Cu-catalysts generated from ZnII single-sites: Enhanced CH₃OH selectivity compared to Cu/ZnO/Al₂O₃. *J. Catal.* **394**, 266–272 (2021).
4. Li, M. *et al.* Thermal stability of size-selected copper nanoparticles: Effect of size, support and CO₂ hydrogenation atmosphere. *Appl. Surf. Sci.* **510**, 145439 (2020).
5. Ji, S. *et al.* A Route to Phase Controllable Cu₂ZnSn(S_{1-x}Se_x)₄ Nanocrystals with Tunable Energy Bands. *Sci. Reports* 2013 31 3, 1–7 (2013).

6. Kawai, J., Yamamoto, T., Tohno, S. & Kitajima, Y. Comparison between X-ray photoelectron and X-ray absorption spectra of an environmental aerosol sample measured by synchrotron radiation. *Spectrochim. Acta Part B At. Spectrosc.* **54**, 241–245 (1999).
7. Van Den Berg, R. *et al.* Structure sensitivity of Cu and CuZn catalysts relevant to industrial methanol synthesis. *Nat. Commun.* **2016** 71 **7**, 1–7 (2016).
8. Dalebout, R. *et al.* Insight into the Nature of the ZnO_x Promoter during Methanol Synthesis. *ACS Catal.* **12**, 6628–6639 (2022).
9. Hu, B. *et al.* Cu@ZIF-8 derived inverse ZnO/Cu catalyst with sub-5 nm ZnO for efficient CO₂ hydrogenation to methanol. *Catal. Sci. Technol.* **9**, 2673–2681 (2019).
10. An, B. *et al.* Confinement of Ultrasmall Cu/ZnO_x Nanoparticles in Metal-Organic Frameworks for Selective Methanol Synthesis from Catalytic Hydrogenation of CO₂. *J. Am. Chem. Soc.* **139**, 3834–3840 (2017).
11. Zhang, J. *et al.* Neighboring Zn-Zr Sites in a Metal-Organic Framework for CO₂ Hydrogenation. *J. Am. Chem. Soc.* **143**, 8829–8837 (2021).
12. Kresse, G. & Hafner, J. *Ab initio* molecular dynamics for liquid metals. *Phys. Rev. B* **47**, 558 (1993).
13. Kresse, G. & Furthmüller, J. Efficiency of *ab-initio* total energy calculations for metals and semiconductors using a plane-wave basis set. *Comput. Mater. Sci.* **6**, 15–50 (1996).
14. Kresse, G. & Furthmüller, J. Efficient iterative schemes for *ab initio* total-energy calculations using a plane-wave basis set. *Phys. Rev. B* **54**, 11169–11186 (1996).
15. Perdew, J. P., Burke, K. & Ernzerhof, M. Generalized gradient approximation made simple. *Phys. Rev. Lett.* **77**, 3865–3868 (1996).
16. Zhou, H. *et al.* Engineering the Cu/Mo₂CT_x (MXene) interface to drive CO₂ hydrogenation to methanol. *Nat. Catal.* **2021** 410 **4**, 860–871 (2021).
17. Kim, C. *et al.* Energy-efficient CO₂ hydrogenation with fast response using photoexcitation of CO₂ adsorbed on metal catalysts. *Nat. Commun.* **2018** 91 **9**, 1–8 (2018).
18. Zhang, W. & Xiao, Y. Mechanism of Electrocatalytically Active Precious Metal (Ni, Pd, Pt, and Ru) Complexes in the Graphene Basal Plane for ORR Applications in Novel Fuel Cells. *Energy and Fuels* **34**, 2425–2434 (2020).
19. Ahmad, R. & Singh, A. K. Synergistic core–shell interactions enable ultra-low overpotentials for enhanced CO₂ electro-reduction activity. *J. Mater. Chem. A* **6**, 21120–21130 (2018).
20. Alvarez-Garcia, A., Flórez, E., Moreno, A. & Jimenez-Orozco, C. CO₂ activation on small Cu-Ni and Cu-Pd bimetallic clusters. *Mol. Catal.* **484**, 110733 (2020).
21. Fan, Q. Y., Sun, J. J., Wang, F. & Cheng, J. Adsorption-Induced Liquid-to-Solid Phase Transition of Cu Clusters in Catalytic Dissociation of CO₂. *J. Phys. Chem. Lett.* **11**, 7954–7959 (2020).

REVIEWER COMMENTS

Reviewer #2 (Remarks to the Author):

After the revision, the authors have addressed some of my concerns last time with adding experimental data and additional discussion. Overall, the quality of this work has been promoted. However, I still have some apprehensions.

1. How to prove that Cu nanoparticles were encapsulated in the MOF structure?

2. The author found that Cu nanoparticles encapsulated on the Zn-based MOF that are highly methanol selective (> 90%) and the CH₄ selective is close to zero. However, the obvious characteristic vibration bands of CH₄ (3015 and 1305 cm⁻¹) are detected in situ DRIFTS (Figs. 6b and 6c), indicating the presence of CH₄ in the process.

3. The activity as well as the stability of the reported catalysts for methanol formation is not attractive compared with the Cu-based catalysts in the reported literatures (we can find many related results in the reviews Chem. Rev. 2017, 117, 9804–9838, Chem. Rev. 2020, 120, 7984–8034) under the similar conditions (H₂/CO₂ = 4, 5MPa, 525K, shown in Fig. S11). The role of MOF structure in the present work in determining the catalyst performance is still unclear.

4. The response for the revision is unclear and irresponsible. The authors should, at least, show what and where they made the revision, rather than just claiming that they had made revision. The added data and discussions are also strongly suggested to show in the response. In the present version, I spent a lot of time to find the related revisions.

Reviewer #3 (Remarks to the Author):

The revision of the manuscript by Velisoju et al. about the generation of Cu-nanoclusters in ZIF-8 addresses all the comments and suggestions I made. I thank the authors for taking their time to address the issues pointed out in comments 3.2 and 3.3 regarding the inconsistency on pg. 11 regarding the Cu-nanoparticle formation, Fig. 6a on pg. 15, and for enhancing the comparability of their work by adding the TOF and STY of the methanol synthesis with their material.

Moreover, I greatly appreciate the overhaul of the theoretical section of the manuscript. The methodology and evaluation for the description of the catalytic system are sound and sensible, fitting well with the experimental results.

However, I do recommend reviewing the section on the DFT calculations (pg.23-24, starting line 487) regarding its content, as there seems to be doubled information (e.g., it is mentioned twice that VASP is

being used in lines 487/488 and lines 504/505) that could be condensed for better readability. Also, I suggest adding a citation for the Grimme DFT-D3 correction.

Apart from that, I have no further comments to add.

Reviewer #2:

Comment #2.1: After the revision, the authors have addressed some of my concerns last time with adding experimental data and additional discussion. Overall, the quality of this work has been promoted. However, I still have some apprehensions.

§ Response 2.1. We thank the reviewer again for his valuable time and comments in assessing our work in both the revisions that allowed us to improve the quality of the manuscript.

Comment #2.2: How to prove that Cu nanoparticles were encapsulated in the MOF structure?

§ Response 2.2. We presented HRTEM and HAADF-STEM-EDX techniques to confirm the speciation of copper after exchange and reduction procedures (Fig. 2) and after 70 h induction (or aging) of the catalyst (Fig. 5e). We agree with the recommendations made by the referee. Indeed, solely relying on TEM images can make it challenging to establish the presence of copper particles conclusively occluded within the ZIF-8 crystals. However, our analysis of the pre-reduction samples reveals no visible particles (Fig. 2). Combined with other characterizations, such as infrared and XRD (Fig. 1), we demonstrate that copper is uniformly dispersed throughout the crystal.

Upon reduction of the catalysts, we observe the formation of very small copper particles that appear to be encapsulated within the crystals (Fig. 2). DFT results (Fig. 7) also indicate that the particles occluded within the ZIF-8 crystals exhibit distinct electronic properties compared to those not interacting with the framework. When considered alongside the additional characterizations, this provides compelling evidence of the significance of these occluded particles in achieving desirable catalytic properties.

Moreover, when we compare the results of the sample prepared via ion exchange (Cu/ZIF-8|IE|) with the sample prepared through impregnation (Cu/ZIF-8|IM|), where the copper particles were located externally (Fig. S4 and Fig. S8), it becomes evident that the Cu/ZIF-8|IE| sample exhibits significantly higher activity and selectivity. These findings powerfully point to the assertion that the particles are housed within the crystals.

Comment #2.3: The author found that Cu nanoparticles encapsulated on the Zn-based MOF are highly methanol selective (>90%) and the CH₄ selective is close to zero. However, the apparent characteristic vibration bands of CH₄ (3015 and 1305 cm⁻¹) are detected in situ DRIFTS (Figs. 6b and 6c), indicating the presence of CH₄ in the process.

§ Response 2.3. We wish to clarify that Figure 6b pertains to the spectra of bare ZIF-8, copper-exchanged, and impregnated ZIF-8 samples without the adsorption of reactants. The figure caption has been accordingly modified on p15. Furthermore, no vibrational band is observed at 3015 cm⁻¹ in Fig. 6c, and the vibrational band present at around 3023 cm⁻¹ is due to ZIF-8 itself, as the same band is also observed in the case of catalyst before adsorption of any reactants.

Comment #2.4: The activity as well as the stability of the reported catalysts for methanol formation is not attractive compared with the Cu-based catalysts in the reported literature (we can find many related results in the reviews Chem. Rev. 2017, 117, 9804–9838, Chem. Rev. 2020, 120, 7984–8034) under the similar conditions (H₂/CO₂ = 4, 5 MPa, 525K, shown in Fig. S11). The role of MOF structure in the present work in determining the catalyst performance is still unclear.

§ Response 2.4. We agree with the comment that copper and copper-zinc-based catalytic systems are reported under similar conditions using Cu supported on mixed-metal oxides. Nonetheless, in this context, we investigate an alternative category of copper-based systems that leverage MOFs as a support. This approach addresses the challenges associated with methanol selectivity, a common issue encountered in traditional copper-based catalysts, including commercial Cu/ZnO/Al₂O₃. In this context, we report a Zn-based MOF with copper for the first time without altering the MOF network with finely dispersed copper and higher stability under methanol synthesis conditions. Based on the suggestions and literature in the field, we provide a standard and most stringent comparison (Table S4) with standard methanol synthesis catalyst Cu/ZnO/Al₂O₃ and similar MOF-based works to address the selectivity and deactivation issues. As presented in the manuscript, our catalyst outperformed Cu/ZnO/Al₂O₃ and other reported MOFs regarding space-time yields and TOF with outstanding stability under industrially relevant methanol synthesis conditions. This last aspect is original in the literature.

Regarding the role of MOF structure, we chose this ZIF-8 MOF as it is commercially available, Zn-based, and possesses good thermal stability, high porosity, surface area, and ordered structure. This allowed us to finely disperse copper particles with uniform and small particle size distribution to avoid selectivity issues, thereby deactivating by sintering. As we confirm with different techniques (XRD, TGA, N₂-physisorption, and FTIR), the structure of the MOF remains unaltered during copper exchange, pre-treatment, and finally during the reaction. Additionally, as we prove from the experiments and the DFT calculations, the structure of ZIF-8 allowed copper species to stabilize during the exchange, reduction, and reaction procedure and remain unaltered even after 150 h of reaction time.

Comment #2.5: The response for the revision is unclear and irresponsible. The authors should, at least, show what and where they made the revision, rather than just claiming that they had made revision. The added data and discussions are also strongly suggested to show in the response. In the present version, I spent a lot of time to find the related revisions.

§ Response 2.5. We are grateful for your comments and regret that our previous response may have seemed unclear. We have considered all the suggestions and comments you and the other reviewers made, and we apologize for any confusion if this was not perceived as such. We have strived to be more explicit and transparent in this new version of our revisions.

Reviewer #3:

Comment #3.1: The revision of the manuscript by Velisoju et al. about the generation of Cu-nanoclusters in ZIF-8 addresses all the comments and suggestions I made. I thank the authors for taking their time to address the issues pointed out in comments 3.2 and 3.3 regarding the inconsistency on pg. 11 regarding the Cu-nanoparticle formation, Fig. 6a on pg. 15, and for enhancing the comparability of their work by adding the TOF and STY of the methanol synthesis with their material. Moreover, I greatly appreciate the overhaul of the theoretical section of the manuscript. The methodology and evaluation for the description of the catalytic system are sound and sensible, fitting well with the experimental results.

§ Response 3.1. We thank the reviewer again for his time and invaluable input in the previous revision that allowed us to improve the quality of the manuscript, especially the TOF calculation, which allowed us to enhance the comparability of our results with the existing ones in the literature.

Comment #3.2: However, I do recommend reviewing the section on the DFT calculations (pg.23-24, starting line 487) regarding its content, as there seems to be doubled information (e.g., it is mentioned twice that VASP is being

used in lines 487/488 and lines 504/505) that could be condensed for better readability. Also, I suggest adding a citation for the Grimme DFT-D3 correction. Apart from that, I have no further comments to add.

§ Response 3.1. Following the comment, we have removed the repeated experimental part and added the relevant reference for the Grimme DFT-D3 correction in the revised manuscript on p 23.

REVIEWERS' COMMENTS

Reviewer #2 (Remarks to the Author):

I have no additional comments on this manuscript.